# Spectral Sizing of a Coarse Spectral Resolution Satellite Sensor for XCO$_2$

Jonas Simon Wilzewski[1,2], Anke Roiger[1], Johan Strandgren[1], Jochen Landgraf[3], Dietrich G. Feist[4,1,5], Voltaire A. Velazco[6], Nicholas M. Deutscher[6], Isamu Morino[7], Hirofumi Ohyama[7], Yao Té[8], Rigel Kivi[9], Thorsten Warneke[10], Justus Notholt[10], Manvendra Dubey[11], Ralf Sussmann[12], Markus Rettinger[12], Frank Hase[13], Kei Shiomi[14], and André Butz[15]

[1]Deutsches Zentrum für Luft- und Raumfahrt, Institut für Physik der Atmosphäre, Oberpfaffenhofen, Germany
[2]Meteorological Institute Munich, Ludwigs-Maximilians-Universität, Munich, Germany
[3]Netherlands Institute for Space Research, Utrecht, Netherlands
[4]Ludwig-Maximilians-Universität München, Lehrstuhl für Physik der Atmosphäre, Munich, Germany
[5]Max Planck Institute for Biogeochemistry, Jena, Germany
[6]Centre for Atmospheric Chemistry, School of Earth, Atmospheric and Life Sciences, University of Wollongong, NSW, Australia
[7]National Institute for Environmental Studies (NIES), Tsukuba, Japan
[8]LERMA-IPSL, Sorbonne Université, CNRS, Observatoire de Paris, Université PSL, 75005, Paris, France
[9]Finnish Meteorological Institute, FMI, Sodankylä, Finland
[10]Institute of Environmental Physics, University of Bremen, Bremen, Germany
[11]Earth System Observations, Los Alamos National Laboratory, Los Alamos, NM 87545, USA
[12]Karlsruhe Institute of Technology, IMK-IFU, Garmisch-Partenkirchen, Germany
[13]Karlsruhe Institute of Technology, IMK-ASF, Karlsruhe, Germany
[14]Japan Aerospace Exploration Agency, Tsukuba, Japan
[15]Institute of Environmental Physics, University of Heidelberg, Heidelberg, Germany

**Correspondence:** Jonas Wilzewski (jonas.wilzewski@dlr.de)

**Abstract.**

Verifying anthropogenic carbon dioxide (CO$_2$) emissions globally is essential to inform about the progress of institutional efforts to mitigate man-made climate forcing. To monitor localized emission sources, spectroscopic satellite sensors have been proposed that operate on the CO$_2$ absorption bands in the shortwave-infrared (SWIR) spectral range with ground resolution as fine as a few tens to about a hundred meters. When designing such sensors, fine ground resolution requires a trade-off towards coarse spectral resolution in order to achieve sufficient noise performance. Since fine ground resolution also implies limited ground coverage, such sensors are envisioned to fly in fleets of satellites, requiring low-cost and simple design, e.g. by restricting the spectrometer to a single spectral band.

Here, we use measurements of the Greenhouse Gases Observing Satellite (GOSAT) to evaluate the spectral resolution and spectral band selection of a prospective satellite sensor with fine ground resolution. To this end, we degrade GOSAT SWIR spectra of the CO$_2$ bands at 1.6 (SWIR-1) and 2.0 $\mu$m (SWIR-2) to coarse spectral resolution, without a further addition of noise, and we evaluate single-band retrievals of the column-averaged dry-air mole-fractions of CO$_2$ (XCO$_2$) by comparison to ground-truth provided by the Total Carbon Column Observing Network (TCCON) and by comparison to global "native" GOSAT retrievals with native spectral resolution and spectral band selection. Coarsening spectral resolution from GOSAT's

native resolving power of >20,000 to the range of 700 to a few thousand makes the scatter of differences between the SWIR-1 and SWIR-2 retrievals and TCCON increase moderately. For resolving powers of 1,200 (SWIR-1) and 1,600 (SWIR-2), the scatter increases from 2.4 ppm (native) to 3.0 ppm for SWIR-1 and 3.3 ppm for SWIR-2. Coarser spectral resolution yields only marginally worse performance than the native GOSAT configuration in terms of station-to-station variability and geophysical parameter correlations for the GOSAT-TCCON differences. Comparing the SWIR-1 and SWIR-2 configurations to native GOSAT retrievals on the global scale, however, reveals that the coarse resolution SWIR-1 and SWIR-2 configurations suffer from some spurious correlations with geophysical parameters that characterize the light-scattering properties of the scene such as particle amount, size, height and surface albedo. Overall, the SWIR-1 and SWIR-2 configurations with resolving powers of 1,200 and 1,600 show promising performance for future sensor design in terms of random error sources while residual errors induced by light-scattering along the lightpath need to be investigated further. Due to the stronger $CO_2$ absorption bands in SWIR-2 than in SWIR-1, the former has the advantage that measurement noise propagates less into the retrieved $XCO_2$ and that some retrieval information on particle scattering properties is accessible.

## 1 Introduction

Accurate and spatiotemporally densely resolved information on localized carbon dioxide ($CO_2$) emission sources such as power plants is crucial to inform about $CO_2$ emission reduction targets that national, regional, and municipal administrations worldwide have committed to through their climate action plans. Satellite remote sensing of the column-averaged dry-air mole fractions of $CO_2$ ($XCO_2$) could contribute to providing such crucial information if satellite design succeeds in combining fine ground resolution with sufficient precision and if satellite concepts are simple enough to allow for a fleet of sensors enabling broad coverage of the globe.

Global $XCO_2$ concentration measurements from space were pioneered by the SCanning Imaging Absorption SpectroMeter for Atmospheric CHartographY mission, SCIAMACHY (e.g. Burrows et al., 1995; Reuter et al., 2010; Schneising et al., 2013), with ground resolution of ~60×30 km$^2$ (Bovensmann et al., 1999). Finer ground resolution (with sparse sampling, though) was subsequently achieved by the Greenhouse Gases Observing Satellite (GOSAT, 10.5 km diameter ground footprint) (Kuze et al., 2009, 2016) and the Orbiting Carbon Observatory (OCO-2, 1.3×2.3 km$^2$ ground footprint) (Crisp et al., 2008, 2017). The Chinese TanSat mission has also embarked on this strategy (Yang et al., 2018). GOSAT and OCO-2 offer insights into the natural processes of the carbon cycle (Guerlet et al., 2013a; Parazoo et al., 2013; Liu et al., 2017; Chatterjee et al., 2017) as well as into anthropogenic emission patterns (Hakkarainen et al., 2016). Urban carbon dioxide signals have been detected by these instruments, for example in the Los Angeles basin (Kort et al., 2012; Eldering et al., 2017; Schwandner et al., 2017). Nassar et al. (2017) have demonstrated the ability of OCO-2 to observe anthropogenic $CO_2$ emissions from individual, coal-fired power plants showcasing the added value of imaging information. A similar concept has been put forward by the CarbonSat

mission (Bovensmann et al., 2010), which has evolved into a candidate for a future European carbon monitoring mission (e.g. Pillai et al., 2016; Broquet et al., 2018; Reuter et al., 2019). The CO2M mission currently under investigation at the European Space Agency aims at ground resolution of 4 km$^2$ (Sierk et al., 2019; Wu et al., 2019a). All these satellite missions and concepts rely on a multi-band spectral configuration that covers the oxygen (O$_2$) A-band at roughly 0.76 $\mu$m (NIR), and

the CO$_2$ bands at 1.6 (SWIR-1) and 2.0 $\mu$m (SWIR-2). The spectral resolution ranges from resolving powers $\frac{\lambda}{\Delta\lambda} > 20,000$ (with $\lambda$ the wavelength and $\Delta\lambda$ the full-width-half-maximum of the instrument spectral response function) for GOSAT, OCO-2, and Tansat to $\frac{\lambda}{\Delta\lambda} > 6,000$ for CO2M's SWIR-2 band and $\frac{\lambda}{\Delta\lambda} > 4,000$ for CarbonSat's SWIR-2 band. The typical XCO$_2$ native GOSAT retrievals attempt to make use of these bands by retrieving XCO$_2$ simultaneously with atmospheric scattering properties.

For methane (CH$_4$), which poses similar remote sensing challenges as CO$_2$, it has been demonstrated that a satellite spectrometer operating at coarse spectral resolution ($\frac{\lambda}{\Delta\lambda}$ of a few hundred) on a single absorption band (around 2.35 $\mu$m) can achieve successful CH$_4$ hot-spot detection with a ground resolution of 30 m (Thompson et al., 2016). Similar results for CH$_4$ have been reported from aircraft sensors that reach ground pixel sizes on the order of 1-10 m (Dennison et al., 2013; Thorpe et al., 2016a, b; Krings et al., 2018). Dennison et al. (2013) suggested that measuring the 2.0 $\mu$m CO$_2$ bands with a spectral res-

olution of 10 nm ($\frac{\lambda}{\Delta\lambda} \approx 200$) enables a space-borne spectrometer design that results in ground resolutions as fine as $60\times60$ m$^2$. Thorpe et al. (2016a) have shown that their airborne AVIRIS-NG instrument exploiting the CO$_2$ absorption bands at 2.0 $\mu$m at a spectral resolution of roughly 5 nm ($\frac{\lambda}{\Delta\lambda} \approx 400$) enables quantitative retrievals of CO$_2$ in localized emission plumes. Thorpe et al. (2016b) suggested that, for CH$_4$, a spectrometer design with a spectral resolution of 1 nm ($\frac{\lambda}{\Delta\lambda} \approx 2,000$) could provide an optimal trade-off that allows for accurate CH$_4$ quantification while supporting small ground pixels.

This study is motivated by the margins that coarse spectral resolution offers with respect to improving ground resolution and that single-band configurations offer with respect to deploying a fleet of several low-cost satellites. Fig. 1 schematically illustrates the key advantage of an assumed $50\times50$ m$^2$ ground resolution spectrometer over an instrument with km-scale resolution for point-source observation. If the localized source plume does not fill the satellite's entire ground pixel, the XCO$_2$ enhancement averages with the background concentration field over the satellite pixel. For the example in Fig. 1, this leads to

a maximum of 3 ppm enhancement for a satellite sensor with $2\times2$ km$^2$ ground resolution. Shrinking the ground pixels leads to larger enhancements in the vicinity of the source, simply because the plume fills a larger portion of the (smaller) pixels. In Fig. 1, $50\times50$ m$^2$ ground resolution delivers 12 ppm enhancement at 2 km downwind distance, plus a sampling of the plume cross-section by more than 10 pixels. Further downwind, where the plume has laterally spread to the km-scale, enhancements per pixel are similar for fine and coarse ground resolution, but the fine ground resolution sensor would still sample the plume by

multiple ground pixels. Thus, a sensor with fine ground resolution allows for less stringent precision requirements (per ground pixel), and it could potentially resolve plume shapes at some detail. Since small ground pixels imply less backscattered photons, sensor design for fine ground resolution typically needs to compensate by enhancing light throughput of the spectrometer and by collecting more photons in the spectral domain, e.g. by coarsening spectral resolution. Since finer ground resolution implies narrower ground coverage for the same detector size, global monitoring with fine ground resolution almost certainly implies

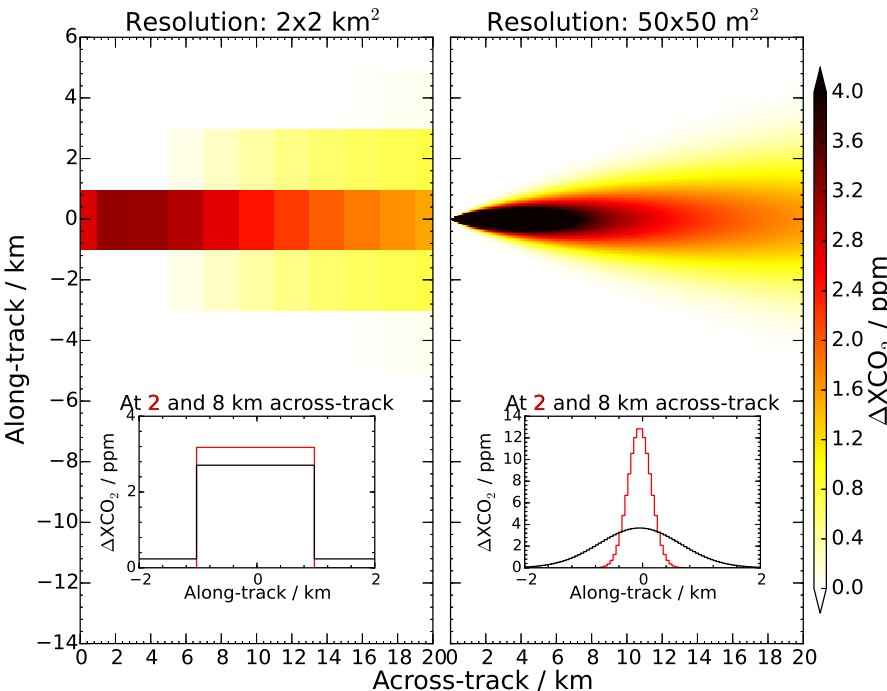

**Figure 1.** Schematic Gaussian plume of the $XCO_2$ enhancement ($\Delta XCO_2$) originating from a power plant with 12.3 Mt $CO_2$ y$^{-1}$ emission rate (wind from left to right, Guifford-Pasquill stability class C; power plant at the origin; satellite assumed to move from bottom to top, sampling left to right) as seen (without noise) by hypothetical satellite spectrometers with $2\times 2$ km$^2$ ground pixels (left), and with $50\times 50$ m$^2$ ground pixels (right). Insets show $\Delta XCO_2$ measured by the sensors at 2 km (red) and 8 km (black) downwind of the source along the plume cross section (note different $\Delta XCO_2$ scales in insets).

the need for a fleet of sensors which would be easier to realize if the sensors had a simple, single-band configuration instead of full spectral coverage from the NIR into SWIR-2.

Here, we aim at evaluating the performance of a hypothetical $XCO_2$ sensor that has coarse spectral resolution in a single-band configuration. That is, we evaluate a sensor concept which measures the $CO_2$ bands near either 1.6 (SWIR-1) or 2.0 $\mu$m (SWIR-2) with resolving power in the range of 700 to a few thousand, i.e. roughly between the AVIRIS-NG and CarbonSat concepts. Galli et al. (2014) conducted a related study where they spectrally degraded GOSAT soundings to resolutions ranging from native GOSAT resolution down to $\frac{\lambda}{\Delta\lambda} \approx 3,000$ while leaving the multi-band configuration (NIR, SWIR-1, SWIR-2) of the $XCO_2$ retrievals untouched. They found that coarser spectral resolution typically implies larger statistical and systematic $XCO_2$ errors when compared to ground truth. Galli et al. (2014), however, did not address the range of resolving powers and the single-band selection covered here. Recently, Wu et al. (2019b) showed that at OCO's native resolving power of $> 20,000$ a single-band retrieval configuration results in almost unchanged XCO2 retrieval accuracy and precision.

Section 2 explains our methodological approach that spectrally degrades GOSAT measurements of the SWIR-1 or SWIR-2 bands to coarser spectral resolution. In section 3, we assess retrieval performance for the SWIR-1 and SWIR-2 configurations

for various resolving powers by comparing our results to ground-truth from the Total Carbon Column Observing Network (TCCON). Thereby, we derive a target spectral resolution for which we carry out a global evaluation with respect to native GOSAT measurements in section 4. Section 5 discusses and concludes on the findings.

## 2 Methodology

GOSAT measures spectra of backscattered solar radiation in three spectral bands centered on the $O_2$ A-band (NIR), the relatively weak $CO_2$ and $CH_4$ bands in the vicinity of 1.6 $\mu$m (SWIR-1), and the strong $CO_2$ and water vapor ($H_2O$) bands around 2.0 $\mu$m (SWIR-2). GOSAT's thermal infrared band recording telluric emission spectra is not used here. We use the level 1B (L1B) data version 201.202, and we add the two measured polarization directions to represent the backscattered radiances. Due to computational costs, we restrict our analysis to cloud-free, quality screened soundings over land as identified by the

native GOSAT retrievals of the RemoTeC algorithm (Butz et al., 2011) within the Climate Change Initiative of the European Space Agency (ESA) (Buchwitz et al., 2017), available for download at http://www.esa-ghg-cci.org. In total, the set comprises 469,689 L1B spectra in the period from April 1, 2009 to December 31, 2016. A typical GOSAT spectrum together with the coarse resolution variants discussed below is shown in Fig. 2.

A key advantage of GOSAT measurements over other $CO_2$ missions, such as OCO-2, is the wide spectral coverage in SWIR-

1 and SWIR-2. The broad spectral coverage allows for conveniently sizing the retrieval windows without being limited by the actual bandpass of the spectrometer. In particular, GOSAT's SWIR-1 and SWIR-2 bands cover, respectively, two and three rotational-vibrational absorption bands of $CO_2$. In order to mimic a coarse resolution sensor, we convolve the native GOSAT L1B spectra by a Gaussian function of selectable full-width-at-half-maximum (FWHM). Since we want to isolate the effects of spectral resolution and spectral band selection, we do not add extra noise to the convolved spectra, i.e. the level of noise

is determined by the convolution of the noise of the native GOSAT spectra with the coarse resolution Gaussian line shape function. One would expect extra noise when going to smaller ground pixels as we envision for a future sensor. Estimating the extra noise, however, would require a detailed instrument model which is not available here. Our approach essentially relates to conditions under which the detector noise is negligible as typical for GOSAT. Under such conditions, other sources of error can be addressed e.g. through evaluating geophysical parameter correlations (section 3 and 4). A forthcoming study

will discuss noise performance and retrieval simulations for a hypothetical instrument design. Figure 2 illustrates the spectral convolution approach for a hypothetical spectral resolving power of 1,200 (blue line) and 1,600 (red line) in SWIR-1 and SWIR-2, respectively, in comparison to native GOSAT spectra. We assume that the proposed sensor will have a detector with 256 spectral pixels.

The native and degraded GOSAT measurements are submitted to the RemoTeC retrieval algorithm (Butz et al., 2009, 2011;

Guerlet et al., 2013b), which is in routine use for retrieving $XCO_2$ (and $XCH_4$ – throughout this work X*molecule* refers to the column-averaged dry-air mole fraction of a molecule) from GOSAT (Buchwitz et al., 2017), $XCO_2$ from OCO-2 (Wu et al., 2018) and $XCH_4$ from Sentinel-5 Precursor/TROPOMI (Hu et al., 2018). For GOSAT measurements with native spectral resolution, we deploy RemoTeC in its full-physics ("native" GOSAT) mode, i.e. RemoTeC uses four spectral windows within the

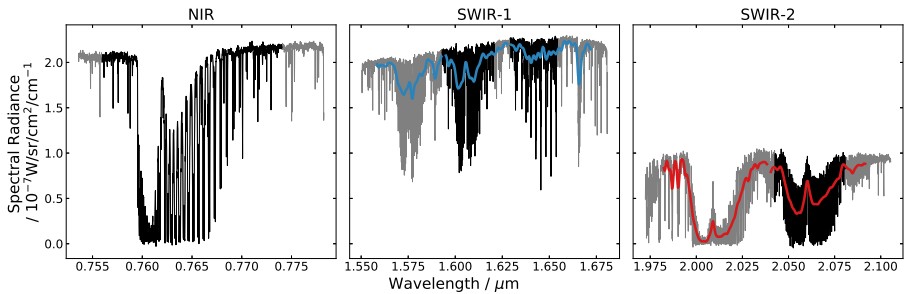

**Figure 2.** Measured GOSAT spectrum of the backscattered radiance in the NIR, SWIR-1 and SWIR-2 (left to right) ranges shown in grey with respective GOSAT retrieval windows in bold black. The spectrally degraded measurements at resolving powers of $\sim 1,200$ (SWIR-1) and $\sim 1,600$ (SWIR-2) are shown in bold blue and bold red respectively.

NIR, SWIR-1 and SWIR-2 ranges (see Table 1 and Fig. 2) and retrieves $XCO_2$, $XCH_4$ together with three particle scattering parameters and other parameters such as surface albedo and spectral shifts. The three particle parameters are the total column number density $N_{par}$, the center height $z_{par}$ of a Gaussian height distribution and the power $\alpha_{par}$ of a power-law size distribution $n(r) \sim r^{-\alpha_{par}}$ with particle radius $r$. The native GOSAT configuration is equivalent to the standard retrieval also in operation for ESA's climate change initiative (e.g. Buchwitz et al. (2017)).

| | Coarse spectral resolution sensor | | native GOSAT |
|---|---|---|---|
| | SWIR-1 | SWIR-2 | |
| | | | 0.7741 - 0.7560 |
| Spectral      Windows Used / nm | 1.559 - 1.593 1.595 - 1.628 1.630- 1.672 | | 1.593 - 1.621 1.629 - 1.654 |
| | | 1.982 - 2.038 2.040 - 2.092 | 2.042 - 2.081 |
| FWHM / cm$^{-1}$ | 0.75 …**5.1** …8.0 | 0.75 …**3.1** …7.0 | 0.24 |
| FWHM / nm | 0.20 …**1.37** …2.15 | 0.31 …**1.29** …2.90 | 0.1 |
| approx. Resolving Power | 8,100 …**1,200** …760 | 6,500 …**1,600** …700 | > 20,000 |

**Table 1.** Spectral windows for the various retrieval configurations. Bold numbers indicate the spectral resolution that was chosen for subsequent analyses (see section 3).

For degraded spectral resolution, we use either SWIR-1 or SWIR-2 alone (see Table 1), from which we retrieve $XCO_2$ (as well as $XCH_4$ in SWIR-1) and auxiliary surface albedo and spectral shift parameters. The spectral degradation of the modeled spectra to coarse resolution follows the same approach as for the measurements. First, RemoTeC calculates spectra for GOSAT's native spectral resolution, then the convolution with a Gaussian function simulates the hypothetical measurements

at coarse spectral resolution. For degraded spectral resolution, the SWIR-1 retrievals also adjust $XH_2O$ and $XCH_4$, but neglect scattering by particles (Rayleigh scattering is included) and thus, no particle scattering parameters are retrieved. This approach, which is essentially a transmittance calculation along the geometric lightpath, is hereafter referred to as non-scattering retrieval. Sensitivity studies have shown that retrieving atmospheric scattering parameters from the individual $CO_2$ bands at coarse spec-

tral resolution in the SWIR-1 band suffers from low information content and results in worse XCO2 retrieval performance than under the non-scattering assumption. In the SWIR-2, we retrieve $XH_2O$ along with $XCO_2$. Employing the standard RemoTeC Phillipps-Tikhonov (e.g. Butz et al., 2012) regularization, we additionally retrieve our standard three particle parameters from SWIR-2. We found a regularization strength that allows for retrieving an average of 0.38 degrees of freedom (DFS) for particles (DFS $\gtrsim 1.5$ are typically found in native GOSAT retrievals). Despite this low DFS, the performance of the retrieval was

significantly improved in comparison to a non-scattering retrieval. As the spectral resolution coarsens, the average degrees of freedom for particles decrease from 0.45 (at 6,500 resolving power) to 0.32 (at 700 resolving power). Although variations in DFS may lead to changes in the ability of the retrieval algorithm to converge towards the minimum of the cost function, more than 75 % of all retrievals converge at any given FWHM that we consider in this study. We note that while we divide the SWIR-1 and SWIR-2 retrievals into several sub-windows, the retrieved $XCO_2$ is coupled among the sub-windows.

The actual spectral retrieval windows are defined in Table 1 and illustrated in Fig. 2. The spectral boundaries of the retrieval windows are identical at all selected FWHM. For the coarse spectral resolution SWIR setups, we have chosen to cover two $CO_2$ absorption bands each, while the native GOSAT retrievals cover only one of the bands in SWIR-1 and one of the bands in SWIR-2. Our choice of spectral retrieval windows maximizes the information on $CO_2$ for the coarse resolution retrievals. However, a fine-tuning of the spectral windows for the proposed sensor may be conducted in a future study with an instrument

noise model at hand. For native GOSAT resolution, the two extra bands would provide mostly redundant information while adding significant computational cost. Further, the coarse spectral resolution configurations cover (almost) transparent ranges in the vicinity of the absorption bands in order to constrain surface albedo, even at coarse spectral resolution. If the spectral boundaries of the retrieval windows lie within the $CO_2$ absorption bands, i.e. parts of the $CO_2$ absorption bands are "cut-off", this loss of information generally leads to poorer retrieval performance with respect to TCCON (not shown here).

For both, native GOSAT and degraded SWIR configurations, airmass information is derived from ECMWF surface pressure reanalyses (ERA-Interim) and topographic data from the Shuttle Radar Tomography Mission (SRTM). For each sounding, we use ECMWF and SRTM data to calculate the ground-pixel average surface pressure and the corresponding dry airmass. This is the standard operation procedure for RemoTeC trace gas retrievals from the GOSAT, OCO-2 and TROPOMI satellite instruments. Errors in the calculation of the airmass can be caused by erroneous satellite pointing; these errors are part of the

overall errors reported for the TCCON validation sites (section 3).

Butz et al. (2013) have shown that the $CO_2$ absorption cross sections used in RemoTeC for the SWIR-1 bands and the $CO_2$ band centered at 2.06 $\mu$m in SWIR-2 are consistent to within 0.16 % while the band centered at 2.01 $\mu$m in SWIR-2 is inconsistent with its neighboring SWIR-2 band. Since Butz et al. (2013) used a shorter measurement period than here, we repeat that study for our period and, we determine a scaling factor for the absorption cross sections at 2.01 $\mu$m with respect

to the 2.06 $\mu$m band. To this end, we select ocean-glint scenes that are confidently free of cloud and aerosol using the "upper-

edge" method (Butz et al., 2013). Then, we run RemoTeC retrievals on the 2.01 $\mu$m and the 2.06 $\mu$m bands separately under the non-scattering assumption. The average ratio of the retrieved $XCO_2$ is our scaling factor, which amounts to 0.981 at native GOSAT spectral resolution (i.e. cross sections of the 2.01 $\mu$m band need to be scaled by 0.981). The "upper-edge" method is also used to adjust the scaling factor at each spectral degradation to reflect the impact of the convolution procedure on the low

resolution spectra. The updated factors differ on the sub-permil level from the correction at native spectral resolution.

## 3    Validation with the TCCON Network

As detailed in section 2, we run $XCO_2$ retrievals for the native GOSAT configuration, and for the coarse spectral resolution SWIR configurations on a global set of cloud-free GOSAT measurements. The SWIR-1 and SWIR-2 configurations are run for various spectral resolutions, i.e. for various values of the FWHM of the Gaussian function that convolves the native GOSAT

spectra. The native GOSAT configuration serves as the reference run corresponding to state-of-the-art full-physics retrievals from a spectrometer with fine spectral resolution and wide spectral coverage (from NIR to SWIR-2). The SWIR-1 and SWIR-2 configurations represent our test cases for a potential future sensor with coarse spectral resolution and single-band spectral coverage. To evaluate our retrievals, we compare retrieved $XCO_2$ with measurements by the ground-based TCCON network (Wunch et al., 2011a, b; Messerschmidt et al., 2011; Kiel et al., 2019) (the stations we do not use could not be colocated

with satellite measurements of our GOSAT dataset). We use data from 24 TCCON stations worldwide from the "GGG2014" dataset (available at https://tccondata.org). GOSAT soundings are defined to be coincident with a TCCON station if the satellite sounding is located within 5° with respect to latitude/longitude of the respective ground station. The GOSAT $XCO_2$ retrieval is then compared to the average of the TCCON $XCO_2$ measurements within $\pm$ 2 hours of the GOSAT sounding time.

   $XCO_2$ precision is commonly quantified through the standard deviation of the differences ("scatter") between GOSAT and

TCCON. Figure 3 shows that, while coarser spectral resolution implies larger scatter overall, there is some margin for the choice of spectral resolution in the SWIR-1 band and the figure suggests that the scatter around TCCON exhibits a "plateau" in resolving power space just beyond the critical spectral resolution necessary to distinguish between two typical adjacent $CO_2$ absorption lines in the SWIR-1 (the critical resolving powers are $\sim$3,300 in SWIR-1 and $\sim$2,700 in SWIR-2). This resolving power is marked by the dotted line in Fig. 3. As spectral lines are blended into a broader spectral shape by our

convolution procedure, the non-scattering SWIR-1 retrieval retains a very similar scatter around TCCON for another 1,000 resolving powers. This pattern is not observed for SWIR-2 scatter around TCCON, which gradually increases towards lower resolving powers (bold red line in Fig. 3). We also conducted a sensitivity study where we switched off the retrieval of particle scattering properties in SWIR-2, i.e. using the same non-scattering configuration in SWIR-2 as in SWIR-1. Then, the scatter of SWIR-2 with respect to around TCCON increases significantly (faint red line in Fig. 3) indicating that while DFS for the

particle retrievals is small, $XCO_2$ retrievals benefit. Our observation that spectral resolution degradation for the SWIR-1 and SWIR-2 configurations generally results in larger scatter (than for the native GOSAT retrievals) is in broad agreement with the tendency reported in both Galli et al. (2014) and Wu et al. (2019a) who, however, did not assess the resolution range reported here.

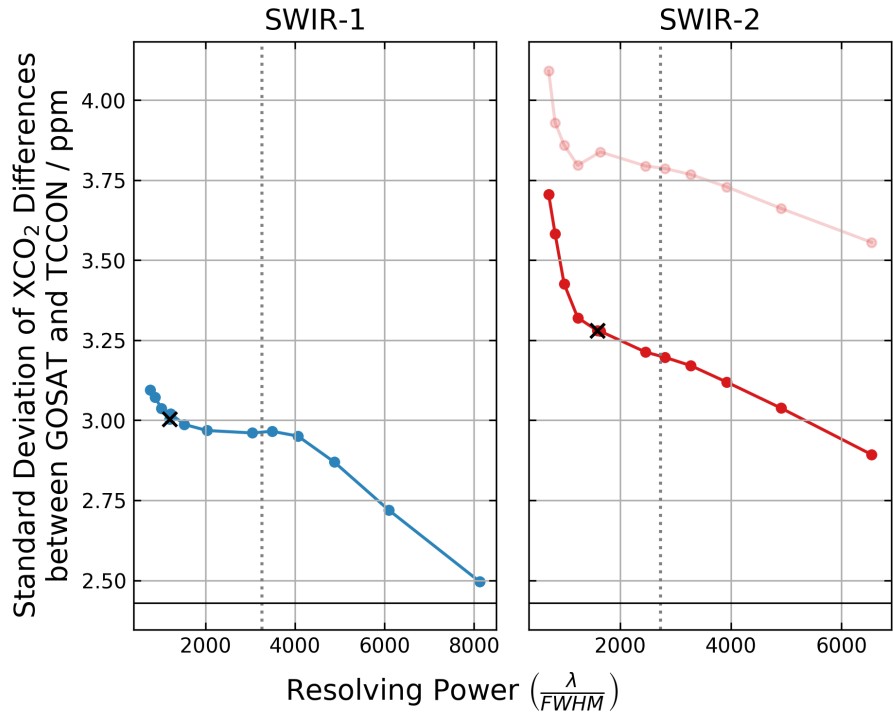

**Figure 3.** Standard deviation of retrieved $XCO_2$ values in SWIR-1 (left) and SWIR-2 (right) around TCCON measurements plotted as a function of resolving power. For SWIR-2, the faint line indicates scatter for a non-scattering retrieval. The dotted line marks the resolving power at which spectral lines become indistinguishable in the convolved spectra. The black horizontal line indicates the native GOSAT scatter around TCCON. The $\times$ marks the resolving power that we study in the rest of the article.

To constrain the resolving power of our future satellite sensor, the scatter around TCCON is the most crucial variable, since the sensor will be built to study local scale $XCO_2$ enhancements. As a consequence, spectral resolving powers greater than the ones that lead to a steep increase in scatter around TCCON in Fig. 3 seem reasonable choices. A technical constraint for the spectral resolution for the envisioned satellite sensor is that the target spectral range ought to be imaged entirely
5   onto the presumed 256 spectral pixels of the sensor's detector assuming a sampling ratio of three. Thereby we define two target resolving powers of 1,200 and 1,600 in SWIR-1 and SWIR-2 (marked with a $\times$ in Fig. 3). For these choices, Figure 4 shows the correlation of the SWIR-1, SWIR-2, and native GOSAT $XCO_2$ retrievals with TCCON. The standard deviations around TCCON amount to 2.43 ppm (native), 3.00 ppm (SWIR-1) and 3.28 ppm (SWIR-2). Given that all retrievals here are without bias correction, the three configurations yield different mean differences ("biases") with respect to TCCON. Generally,
10   although spectral resolution degradation causes a change of the overall bias, a overall bias itself is irrelevant for emission estimates which rely on concentration gradients. Even if the satellite data are to be used in combination with other $CO_2$ measurements, it is common practice to derive a scaling factor of the satellite retrievals with respect to ground-truth.

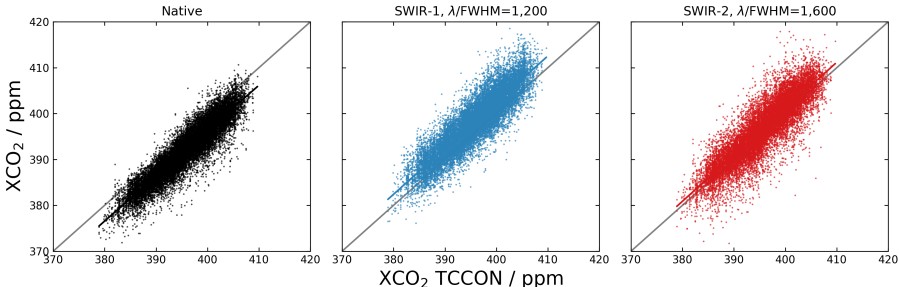

**Figure 4.** Correlation between XCO$_2$ retrieved from GOSAT with the TCCON network. Left: native GOSAT retrieval; Center: SWIR-1 retrieval at 1,200 resolving power; Right: SWIR-2 retrieval at 1,600 resolving power. The grey line indicates a 1:1 correlation line; the colored lines show linear fits to the respective dataset. Standard deviations around TCCON amount to 2.43 ppm (native, compare e.g. Guerlet et al. (2013b)), 3.00 ppm (SWIR-1) and 3.28 ppm (SWIR-2).

Figure 5 resolves the biases per TCCON station for the resolving powers of 1,200 and 1,600 in SWIR-1 and SWIR-2, respectively. Typically, the standard deviation among these station-by-station biases ("bias variability") is taken as a measure for regional systematic errors which cause regional-scale spurious gradients and thus, they are detrimental for regional assessment of sources and sinks. The present retrieval configurations lead to marginally increased TCCON bias variability from 0.94 ppm

for native GOSAT up to 0.99 ppm and 0.97 ppm in SWIR-1 and SWIR-2 retrievals, respectively. Figure 5 also shows XCO$_2$ retrieval standard deviations per TCCON station. The corresponding data for retrieval performance at individual sites can be found in the supplementary materials. Regional scale variability of our proposed retrievals is not of utmost importance as our goal is to make consistent measurements on a local scale. To this end, correlations of retrieval errors caused by parameters that vary on local scales are more informing.

For diagnosing spurious dependencies of the retrieved XCO$_2$ on locally variable geophysical parameters, we examine parameter correlations of the GOSAT-TCCON differences. Fig. 6 shows correlations of the native GOSAT, SWIR-1 (resolving power: 1,200) and SWIR-2 (resolving power: 1,600) retrievals for surface albedo (at 0.774 $\mu$m for native GOSAT, at 1.600 $\mu$m for SWIR-1, at 2.099 $\mu$m for SWIR-2), the scattering optical thickness (SOT) and the three particle parameters $N_{par}$, $z_{par}$, and $\alpha_{par}$ characterizing particle number density, particle layer height and particle size. The particle parameters are taken from na-

tive GOSAT runs since SWIR-1 does not retrieve the parameters and SWIR-2 retrievals exhibit little DFS. The GOSAT-TCCON departures show a small correlation ($R > 0.1$) with surface albedo for both SWIR-1 and native GOSAT configurations, while the SWIR-2 retrievals do not show any correlation. Since the SWIR-1 configuration neglects particle scattering, it appears reasonable that the GOSAT-TCCON departures correlate with albedo which mediates the importance of scattering with respect to the direct lightpath. Yet, only small correlations are found for SWIR-1 errors with SOT, particle layer height and particle

size (R<0.1). Minor SWIR-1 error correlations with respect to particle number density (R=0.11) are present around TCCON stations. For SWIR-2, the correlation with the particle layer height shows $R < -0.3$. Although we do account for scattering in the SWIR-2, the strong regularization of the retrieval leads to convergence close to the a priori ($\tau$=0.1, z$_{par}$=3000 m, $\alpha_{par}$=3.5)

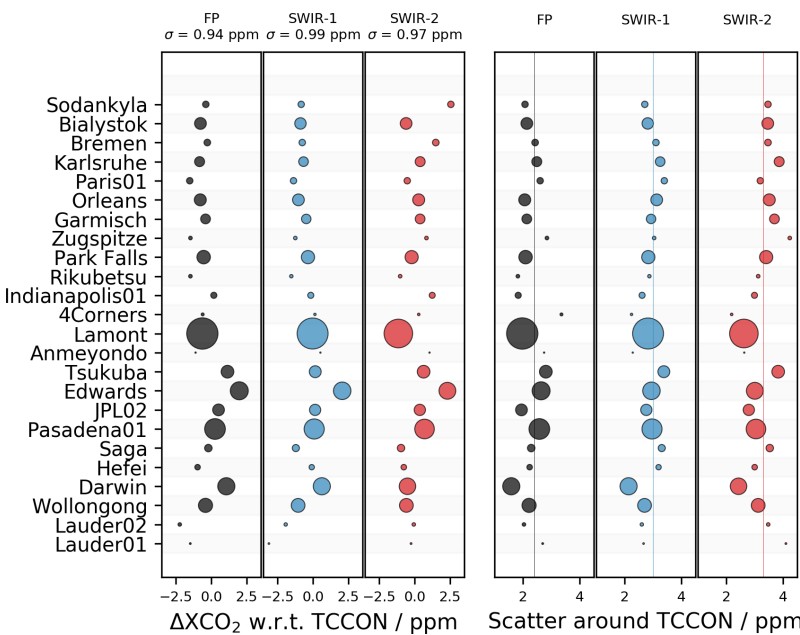

**Figure 5.** Comparison of retrieval performances at individual TCCON stations sorted north to south. Marker size indicates amount of colocated soundings at each station. Left: Station-by-station mean differences between TCCON and the native (black), SWIR-1 (blue), and SWIR-2 (red) retrievals from GOSAT. The standard deviation of mean differences among the stations, $\sigma$, amounts to 0.94 ppm (native), 0.99 ppm (SWIR-1) and 0.97 ppm (SWIR-2). Right: Scatter around TCCON per station for the native, SWIR-1, and SWIR-2 retrievals. Vertical lines mark the average standard deviations (native: 2.43 ppm, SWIR-1: 3.00 ppm, SWIR-2: 3.28 ppm).

of the particle parameters. Therefore, it is not surprising that correlations still exist with particle scattering properties also in SWIR-2. An investigation of the impact of the aerosol priors on retrieval performance showed that SWIR-2 XCO$_2$ is only moderately sensitive to the aerosol priors. For instance, varying aerosol prior optical depth by a factor of two or one half results in small changes in standard deviations around TCCON (+0.22 ppm and −0.08 ppm, respectively). Changing scattering layer

5    height priors to $z_{par}$=1000 m or $z_{par}$=5000 m increased scatter around TCCON by +0.04 ppm and +0.43 ppm, respectively. Similarly, scatter around TCCON changes by +0.22 ppm and −0.05 ppm if $\alpha_{par}$ is set to 3.0 and 5.0, respectively. SWIR-2 retrieval errors around TCCON sites do not significantly correlate with SOT, particle number density and the size parameter. Native GOSAT retrievals consistently show small correlations with all particle parameters. In addition (not shown), correlations with $|R| > 0.1$ are observed for SWIR-1 (and not for SWIR-2) with other geophysical variables like slant airmass of the

10    geometric lightpath (R=-0.17) and water vapor column (R=0.21).

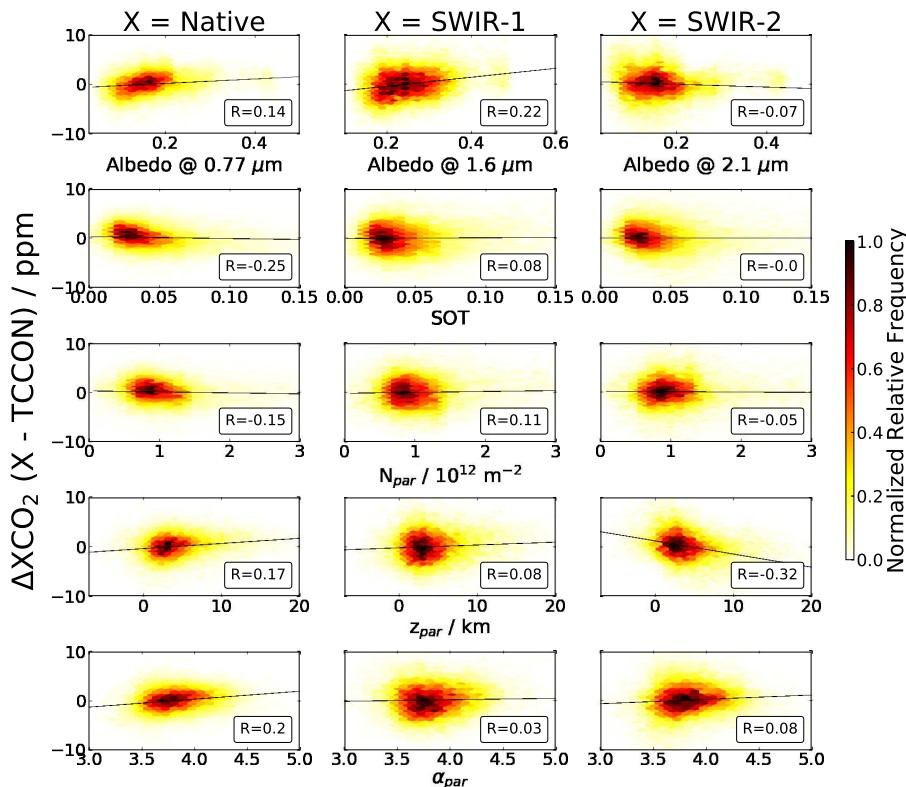

**Figure 6.** Differences between TCCON and native GOSAT (left), SWIR-1 (middle) and SWIR-2 (right) for selected geophysical parameters. Pearson's correlation coefficient $R$ is shown in the corner of each subplot. The solid line is a linear fit to the data. Color encodes relative occurrence of data points.

## 4   Global Evaluation with Native GOSAT Retrievals

For evaluation on the global scale, we take $XCO_2$ from native GOSAT retrievals as the reference. The SWIR-1 and SWIR-2 retrievals are discussed for resolving powers of 1,200 and 1,600, respectively. We subtract the overall biases found by the TCCON analysis from all $XCO_2$ retrievals discussed here (-3.6 ppm, 2.49 ppm and 1.04 ppm for the native, SWIR-1 and SWIR-2 configurations, respectively).

Fig. 7 shows the correlations of the SWIR-1 and SWIR-2 configurations with the native GOSAT retrievals. The standard deviations of the differences to native GOSAT ("scatter") amount to 2.85 ppm and 2.69 ppm for SWIR-1 and SWIR-2, respectively, while correlation coefficients are 0.90 for both SWIR configurations. Although the overall biases with respect to TCCON have been subtracted, the global analysis (containing many more data than the TCCON analysis and even glint spectra) yields non-vanishing mean differences ("bias") of 0.59 ppm for SWIR-1 and -0.29 ppm for SWIR-2 with respect to native GOSAT, presumably as a consequence of an uneven distribution of TCCON sites around the globe. Figure 8 resolves bias and scatter of the SWIR configurations in geographic latitude and season. Figure 8 (upper panels) illustrates that SWIR-1 bias and

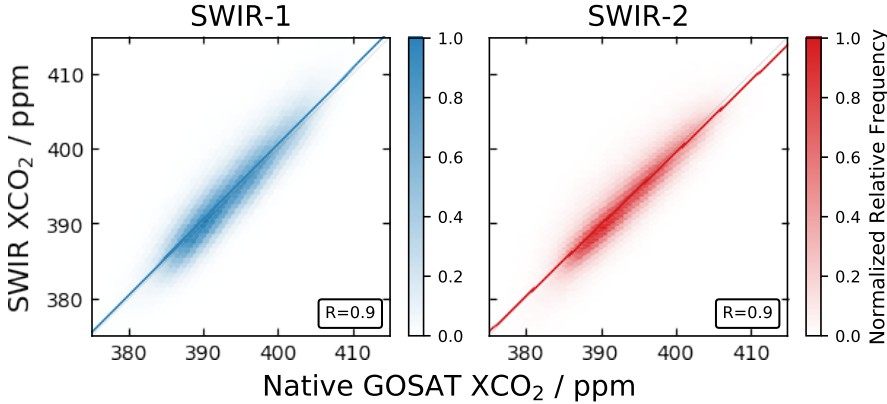

**Figure 7.** Retrieved SWIR-1 (left) and SWIR-2 (right) $XCO_2$ plotted versus the corresponding native GOSAT retrievals. The colored lines indicate linear fits to the data, the grey line marks the 1:1 correlation. Scatter amounts to 2.85 ppm and to 2.69 ppm in SWIR-1 and SWIR-2, respectively. Correlation coefficients are displayed in the lower right corners. Color shading encodes relative occurrence of data points.

scatter are both enhanced in the northern hemisphere. Averaging all seasons, SWIR-1 bias and scatter peak at 1.93 ppm and 3.34 ppm, respectively, between 20 and 30° N where the planet's large deserts are located. Deserts imply bright surfaces and desert dust aerosols which may impact the SWIR-1 retrievals configured under the non-scattering assumption. For the SWIR-2 configuration, figure 8 (lower panels) shows a meridional gradient for scatter and an unclear pattern for bias. Average SWIR-2

scatter varies between 2.03 ppm (at 15° S) and 3.20 ppm (at 65° N). The bias seems to indicate that SWIR-2 retrievals underestimate native GOSAT over the desert latitudes (20° N) and overestimate in higher latitudes (60° N). Seasonal variations generally follow the annual average patterns and no clear seasonal dependencies are detectable. Figures 9 and 10 show maps of the differences between the native GOSAT configuration and SWIR-1 and SWIR-2 averaged on 1×1° for the full record of eight years of GOSAT observations (2009-2016). The global maps retrace the general observations of the zonal averages shown

in Fig. 8. SWIR-1 overestimates native GOSAT retrievals throughout the high albedo regions of the Sahara, central Asia, and tentatively in central Australia. SWIR-2 tends to overestimate native GOSAT in the high latitudes and in Amazonia. Over the deserts the patterns are mixed.

Fig. 11 examines correlations of the retrieval differences with selected geophysical parameters similar to the analysis undertaken for TCCON (section 3, Fig. 6). Among various parameters tested, most significant correlations are found for the

geophysical parameters that control the scattering regime. These parameters are surface albedo, SOT, number density of scatterers ($N_{par}$), center height of the scattering layer ($z_{par}$), the power-law parameter for the scattering particle size distribution ($\alpha_{par}$). As in Fig. 6, particle scattering parameters are taken from the native GOSAT retrievals. Generally, the SWIR-1 and SWIR-2 retrievals show parameter correlations which are more significant on the global scale than what has been found for the TCCON evaluation. The correlation coefficients $R$ are typically on the order of 0.2-0.3 and peak at 0.5 for the correlation

of the SWIR-2 bias with the number density of scatterers $N_{par}$. This is true for the SWIR-2 retrievals although the configura-

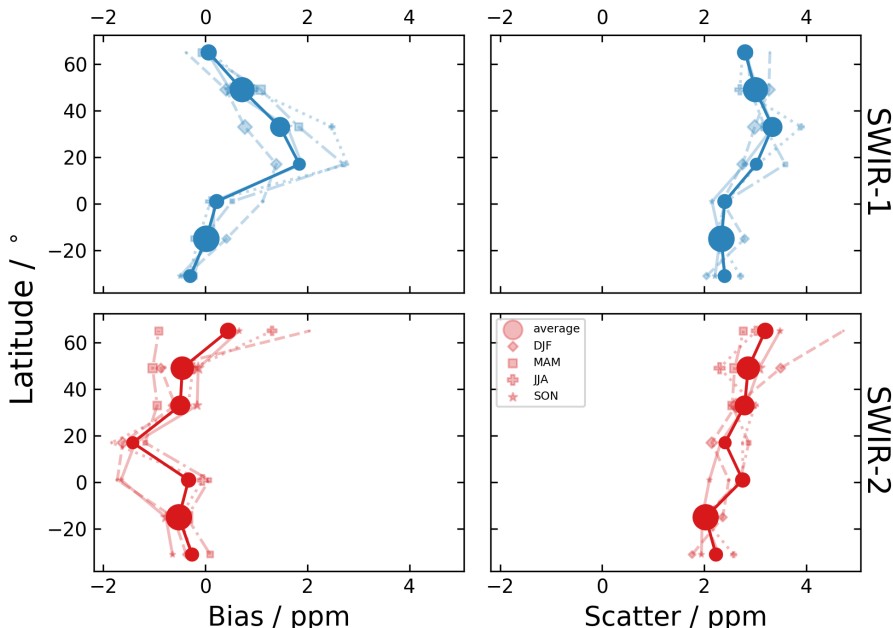

**Figure 8.** Retrieval bias (left) and scatter (right) with respect to native GOSAT $XCO_2$ over land as a function of latitude for the SWIR-1 (top) and SWIR-2 retrievals (bottom) in 16° bins. Bold circles indicate the average bias and scatter, while seasonal variations are shown for boreal winter (DJF, diamonds), spring (MAM, squares), summer (JJA, plus) and fall (SON, stars). Symbol size indicates the relative number of GOSAT observations over land in the respective latitudinal bin.

tion allows for some freedom fitting the particle parameters (DFS=0.38 on average). We also tried a non-scattering variant for SWIR-2 (not shown) which yielded clearly inferior performance (see also figure 3) due to correlations with other parameters such as the water vapor column and the slant airmass.

## 5 Discussion and conclusions

We have evaluated the performance of $XCO_2$ retrievals from solar backscatter satellite observations for a hypothetical sensor that operates at moderate spectral resolution in either the SWIR-1 (around 1.6 $\mu$m) or the SWIR-2 band (around 2.0 $\mu$m). Both configurations, SWIR-1 and SWIR-2, cover tens of $CO_2$ absorption lines and the selected retrieval windows all cover transparent regions toward the shortwave and the longwave ends to constrain surface albedo and its spectral variation. The absorption optical depths in SWIR-1, however, are generally less than those in SWIR-2. SWIR-1, in addition to $CO_2$, covers a $CH_4$ absorption band, both configurations have interfering water vapor absorption. For SWIR-2, we implemented a retrieval variant of the RemoTeC algorithm that allows for estimating three parameters that characterize light scattering in terms of amount, size and height of the scattering particles. Yet, degrees-of-freedom for the particle retrieval amount to only 0.38 on average which indicates that the information content on scattering effects is limited. Nevertheless, our evaluation shows that

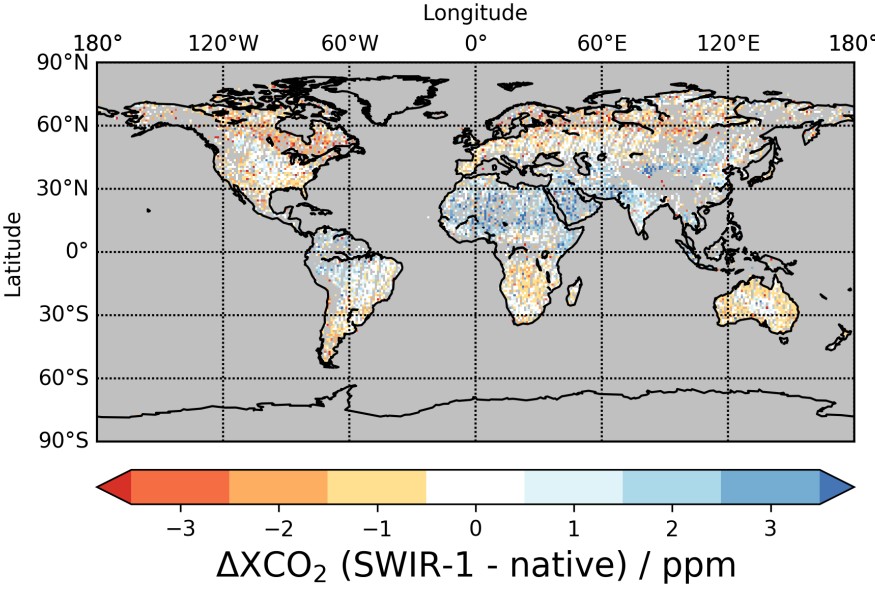

**Figure 9.** Differences between native GOSAT and SWIR-1 retrievals averaged on $1\times1°$ for eight years of GOSAT observations. The global mean bias over land of 0.45 ppm was subtracted from the graph.

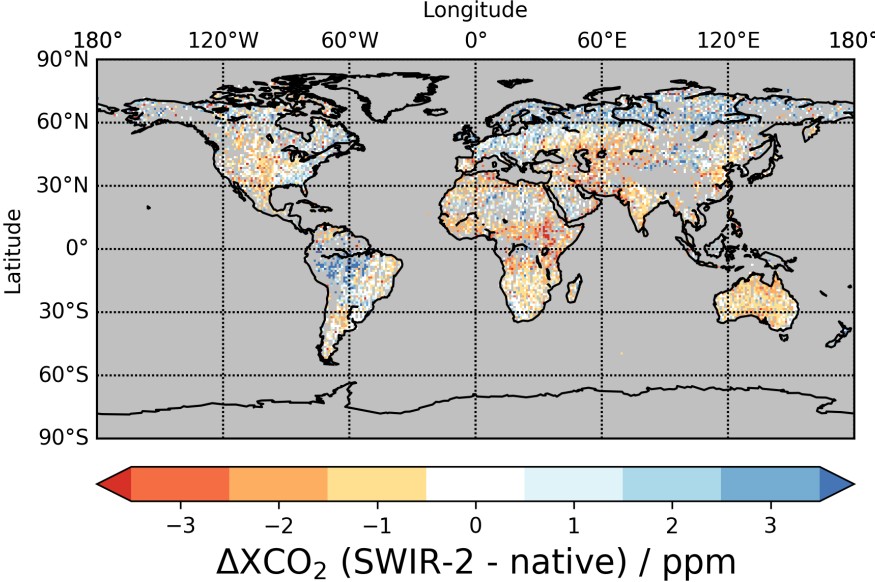

**Figure 10.** Differences between native GOSAT and SWIR-2 retrievals averaged on $1\times1°$ for eight years of GOSAT observations. The global mean bias over land of 0.03 ppm was subtracted from the graph.

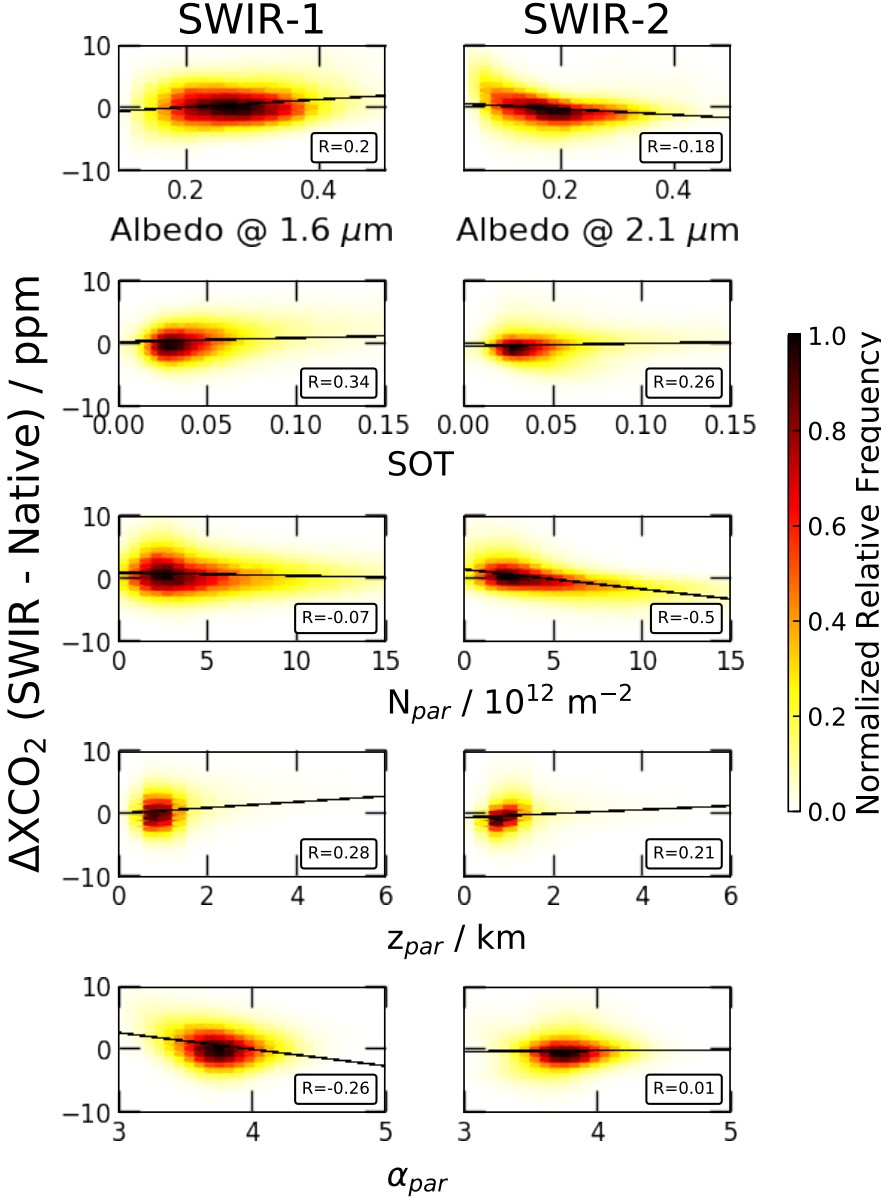

**Figure 11.** Differences between native GOSAT and the SWIR-1 (left) and SWIR-2 (right) configurations for selected geophysical parameters. Pearson's correlation coefficient $R$ is shown in the corner of each subplot. The solid line is a linear fit to the data. Color encodes relative occurrence of data points.

the highly constrained particle retrieval outperforms a non-scattering retrieval in SWIR-2. For SWIR-1, information content on particle scattering was found even less and therefore, the SWIR-1 configuration is based on the non-scattering assumption. Performance was evaluated by mimicking the SWIR-1 and SWIR-2 sensors using spectrally degraded GOSAT observations without further addition of noise to the spectra for the years 2009 to 2016, which allowed for comparisons with ground-truth provided by TCCON and for comparisons with the native GOSAT retrievals (based on GOSAT's full spectral resolution and spectral band coverage).

Comparing the SWIR-1 and SWIR-2 retrievals to TCCON, we tried various resolving powers between 8,100 and 760 for SWIR-1 and 6,500 and 700 for SWIR-2 which is the range roughly in-between the CarbonSat concept and hyperspectral imagers such as AVIRIS-NG. Generally, the scatter with respect to TCCON increases moderately with decreasing resolving power. For SWIR-2, we find a relatively sharp increase below resolving power 1,000. The standard configurations that we have chosen for further analyses correspond to resolving powers of 1,200 and 1,600 for SWIR-1 and SWIR-2, respectively. The corresponding scatter around TCCON amounts to 3.00 ppm and 3.28 ppm, respectively, while native GOSAT retrievals scatter by 2.43 ppm. These configurations fit on a detector with 256 spectral pixels assuming a sampling ratio of 3 per FWHM. Other evaluation metrics such as the overall global bias and the station-by-station biases do not show significantly worse performance than the native GOSAT configuration for the TCCON comparisons. Likewise, correlations with the scattering parameters are mostly small for the TCCON coincidences. The evaluation using the native GOSAT retrievals on the global scale shows differences in the range of 2 to 3 ppm, which, for SWIR-1, clearly correlate with desert areas. In contrast to the TCCON evaluation, the differences to native GOSAT for both, SWIR-1 and SWIR-2, also correlate with particle scattering parameters ($R$ typically in the range 0.2-0.3, up to 0.5). Thus, assuming that the native GOSAT retrievals are more accurate, we expect the SWIR-1 and SWIR-2 configurations to suffer from regionally correlated errors due to particle scattering. Possibly, an additional aerosol sensor, such as the one recently proposed by Hasekamp et al. (2019), may help to overcome challenges in scenes with difficult aerosol loads.

Our goal is to assess the suitability of the spectral sizing of the SWIR-1 and SWIR-2 configurations for a hypothetical sensor that maps localized $CO_2$ sources with high spatial resolution. Our study indicates that limiting band coverage to a single SWIR band and operating at a spectral resolving power between 6,000 and 1,000 does not substantially degrade $XCO_2$ retrieval performance in terms of errors that appear random in our comparisons to TCCON and to the native GOSAT retrievals. However, the SWIR-1 and SWIR-2 configurations are less capable of accounting for particle scattering effects than the configurations of the type of the native GOSAT design. The hypothetical sensor aims at discriminating plumes from background concentration fields on the scale of hundreds of meters to a few kilometers with a ground resolution on the order of $50 \times 50\,\mathrm{m}^2$ enabling enhanced contrast in the vicinity of the sources (Fig. 1). Thus, in terms of random errors, our findings are promising for using one of our SWIR-1 and SWIR-2 configurations. For the errors induced by particle scattering, the implications largely depend on whether the scattering regime can be assumed homogeneous on the respective scales of hundreds of meters to a few kilometers. Even if atmospheric scattering properties are homogeneous, ground albedo varies substantially on these scales. Surface reflectance has been shown to be a central driver in methane retrieval precision by Cusworth et al. (2019). However,

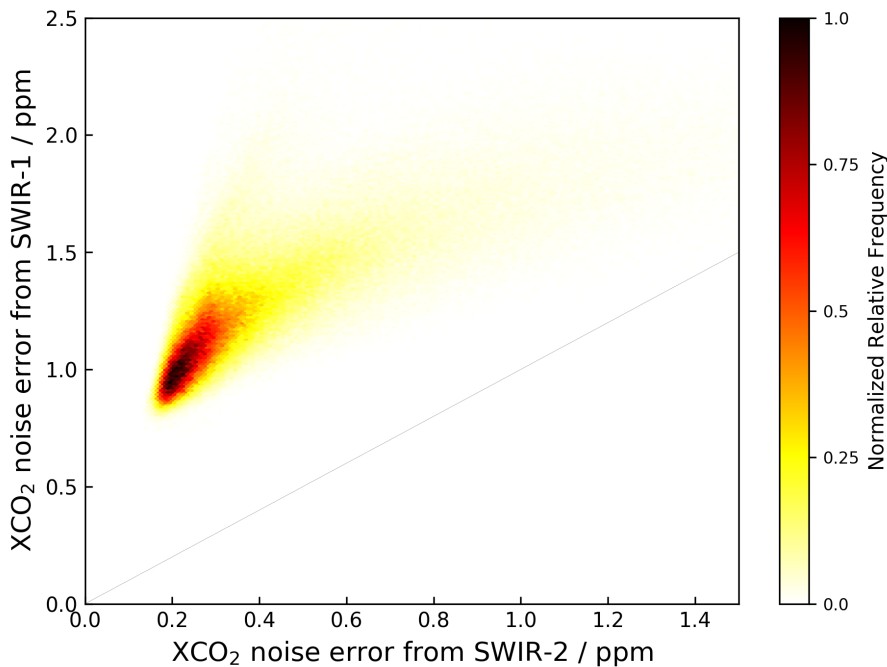

**Figure 12.** $XCO_2$ noise errors for SWIR-1 (ordinate) and SWIR-2 (abscissa). The grey line indicates the 1:1 correlation. Color shading encodes the relative occurrence of data points.

ground albedo is presumably more temporally consistent than aerosols, for example, and so could be more easily defined by independent measurements.

Our study isolates the effects of spectral resolution and spectral band selection, but it postpones the assessment whether sufficient signal-to-noise is achievable. While our evaluation reveals no clear preference for SWIR-1 or SWIR-2, we expect

5   that the assessment of signal-to-noise will favor the SWIR-2 configuration. Fig. 12 shows the noise error of the $XCO_2$ retrievals from SWIR-1 and SWIR-2. These errors are calculated by Gaussian error propagation of GOSAT's radiance noise through the RemoTeC algorithm. Propagated noise errors in SWIR-2 are on average a factor 2.9 less than those in SWIR-1 which is largely due to SWIR-2 covering the stronger $CO_2$ absorption bands. Thus, we expect that achieving the required signal-to-noise is less demanding for SWIR-2 than for SWIR-1. Additionally, the SWIR-2 seems better suited for the construction of a cloud filter,

10   because its $CO_2$ bands have very different optical depths. Similar to the cloud filter currently in use for GOSAT measurements, one could retrieve $XCO_2$ from the two SWIR-2 bands individually and filter for discrepancies. This scheme should be tested in the future. Overall, we recommend further studies to consolidate the SWIR-2 configuration in terms of instrument design and noise performance and to evaluate the relevance of scattering induced errors for the targeted fine spatial resolution. A forthcoming study addressing these aspects of the proposed sensor is currently under preparation.

| TCCON station | Citation | TCCON station | Citation |
|---|---|---|---|
| Sodankyla | Kivi et al. (2014) | Lamont | Wennberg et al. (2016b) |
| Bialystok | Deutscher et al. (2015) | Anmeyondo | Goo et al. (2014) |
| Bremen | Notholt et al. (2014) | Tsukuba | Morino et al. (2018a) |
| Karlsruhe | Hase et al. (2015) | Edwards | Iraci et al. (2016a) |
| Paris | Té et al. (2014) | JPL | Wennberg et al. (2016a) |
| Orleans | Warneke et al. (2014) | Pasadena | Wennberg et al. (2015) |
| Garmisch | Sussmann and Rettinger (2018a) | Saga | Kawakami et al. (2014) |
| Zugspitze | Sussmann and Rettinger (2018b) | Hefei | Liu et al. (2018) |
| Park Falls | Wennberg et al. (2017) | Rikubetsu | Morino et al. (2018b) |
| Izana | Blumenstock et al. (2017) | Ascension Island | Feist et al. (2014) |
| Indianapolis | Iraci et al. (2016b) | Darwin | Griffith et al. (2014a) |
| Four Corners | Dubey et al. (2014), Lindenmaier et al. (2014) | Reunion | De Mazière et al. (2017) |
| Wollongong | Griffith et al. (2014b) | Lauder 1 | Sherlock et al. (2014a) |
| Lauder 2 | Sherlock et al. (2014b) | | |

**Table 2.** Overview of TCCON datasets used in this work.

*Data availability.*

*Author contributions.*

*Competing interests.* The authors declare that they have no conflict of interest.

*Disclaimer.*

*Acknowledgements.* We gratefully acknowledge support from DLR VO-R for funding the young investigator research group "Greenhouse
5  Gases". We thank Luca Bugliaro for valuable feedback on our work. We also thank all TCCON members for making their data available
on the TCCON data archive at https://tccondata.org/. The Ascension Island TCCON station has been supported by the European Space
Agency (ESA) under grant 3-14737 and by the German Bundesministerium für Wirtschaft und Energie (BMWi) under grant 50EE1711C.
The TCCON projects for the Rikubetsu and Tsukuba sites are supported in part by the GOSAT series project. The Paris TCCON site has
received fundings from Sorbonne Université, the French research center CNRS, the French space agency CNES and Région Île-de-France.
10  We acknowledge funding from ESA, DLR and the EC (projects RINGO and VERIFY) for the TCCON sites Bremen, Orleans and Bialystok.
Nicholas Deutscher is supported by an Australian Research Council Future Fellowship, FT180100327. KIT/IMK-ASF (Karlsruhe) and

KIT/IMK-IFU (Garmisch-Partenkirchen, in cooperation with University of Augsburg) acknowledge BMWi for funding data analysis and delivery through DLR projects (Contracts 50EE1711A, B, C, D).

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
