# Peer review of "Spectral Sizing of a Coarse Spectral Resolution Satellite Sensor for $\mathbf{XCO}_2$"

_Atmospheric Measurement Techniques, 2019_

## Referee Comment (RC1) · Anonymous Referee #1 · 3 Sep 2019

This study investigates the performance of a prospective compact carbon dioxide (CO2) satellite sensor with a single-band SWIR spectrometer and with high spatial but moderate spectral resolution. Existing greenhouse gas satellites (GOSAT, OCO-2) rely on a multi-band strategy with high spectral resolution in order to discriminate between signals of the target gas and other interfering signals and to derive critical information on atmospheric scattering e.g. due to aerosols and cirrus clouds.

Downgrading the spectral resolution and using only a single band (e.g. SWIR-2) may thus have significant implications for the accuracy of the measurements. On the other hand, such an instrument could offer substantially increased spatial resolution allowing to image the concentrated CO2 plumes of strong hot-spot emissions.

The idea of trading off spatial resolution for spectral resolution has recently been put

forward in a number of other studies and instrument concepts. The present study is timely and relevant given the growing interest in hot-spot emission detection from space. Its focus on CO2 and a single-band configuration is, to my knowledge, not yet covered in previous studies. The study provides valuable information for the design of such a future instrument especially with respect to the range of spectral resolution/resolving power that is acceptable. The manuscript is very well written and concise. The results are backed up with high quality figures and tables. Overall, I thus recommend publication and only have a few minor points as detailed below:

Minor points: 1. It is unclear to me how column mean dry air mole fractions of CO2 are obtained in a retrieval without NIR band, i.e. in a retrieval where no O2 column is estimated. Where is the information on O2 taken from? From surface pressures from a weather prediction model? How does that add to the overall uncertainty? Isn't the retrieval very sensitive to topographic variations and thus to the pointing accuracy of the instrument in this case?

2. A problem not really addressed in the study is the fact that coarser spectral resolution instruments tend to have larger uncertainties in the spectral calibration. The retrieval can account for spectral shifts, but this is more difficult in case of coarsely resolved spectra. What were the assumptions regarding spectral calibration uncertainties and how would that affect the conclusions?

3. Only quality-screened cloud-free GOSAT spectra were used in the analysis. How much does that screening depend on the information in the NIR and SWIR channels? Or in other words, how much more difficult would quality/cloud screening be for an instrument with a single SWIR channel? This seems important to me, since only a small proportion of pixels usually survive the strict quality flagging required for satellite CO2 retrievals.

Since the main application of the sensor will be point-source detection and quantification, a future study should focus on local rather than global scales as done here. The

recent study of Cusworth et al. (2019; https://doi.org/10.5194/amt-2019-202), for example, shows that local plume detection can be significantly affected by retrieval errors which are correlated with surface reflectance. The spectral resolution of the instrument proposed here may be high enough to mitigate such problems, but this aspect should receive more attention in a future study.

Congratulations, I didn't discover any typos or grammatical errors!

---

## Referee Comment (RC2) · Anonymous Referee #2 · 3 Sep 2019

\<General Comments\>

A space borne imaging spectrometer with 50 m spatial resolution is useful to detect enhancement from CO2 point sources. The feasibility study with real GOSAT data is excellent. The authors are proposing the system with moderate spectral resolution without O2 A band. GOSAT has high spectral resolution O2A band to estimate the light path modification by particles and cloud screening. The authors traded-off the spectral resolution but performance with and without O2A is not clear. Did they consider an O2A spectrometer with moderate spectral resolution? The authors' interest is city observations. Most of the TCCON sites are not located in large cities and aerosol optical thickness over TCCON locations is usually smaller than cities. GOSAT has measured several data over cities such as Tokyo and LA using its target observation function. The

authors can pick up and discuss aerosol effect over cities. They also concluded that dust is the largest error source by specifying latitudinal ranges. GOSAT has observed desert area such as Sahara and Arabian Desert with its medium gain. The authors can analyze directly by picking up medium gain data I recommend major revision for publication.

\<Specific Comments\>

(1) Page 1, Abstract The spectral resolution coarser than native GOSAT and the single-band of CO2 without O2A band are both key parts of this study. However, the latter is not clearly mentioned in the abstract.

(2) page 6, Line 18, "non-scattering retrieval", Page 8, Line 9, "the non-scattering SWIR-1 retrieval" Brief description is needed.

(3) Page 6, Line 20, "More than 75 % of all retrievals converge at any given FWHM that we consider in this study." It is difficult to understand

(4) Page 7, Line 19, "1.856%" It is not clear. Is it 1.8% of XCO2? 7.4 ppm?

(5) Page 10, Figure 4 Use of three individual figures will become clearer.

(6) Page 16, Line 30 "an additional aerosol sensor may help" The largest error source seems to be vertical profile of particles. Conventional aerosol imager provides horizontal distribution only. Which kind of sensor do authors consider?

\<Technical Corrections\>

(1) Page 6, Line 11 XH2O Definition of "XH2O" should be described

(2) Page 12, Figure 12 At present, it is monochromatic. It should be a color figure such as figures 6 and 11. Grey line is difficult to see.

---

## Referee Comment (RC3) · Anonymous Referee #3 · 16 Sep 2019

Review of "Spectral Sizing of a Caorse Spectral Resolution Satellite Sensor for XCO$_2$" by Wilzewski et al, AMTD.

This paper aims to test the fidelity of single-band retrievals of XCO2 at low to moderate spectral resolution, based either on the weak CO$_2$ band only (at 1.6 microns, spectral resolution 1200) or the strong CO$_2$ absorption band (near 2 microns, spectral resolution 1600). The authors do this by applying the RemoTeC retrieval algorithm to both native-resolution, 3-band GOSAT observations, or GOSAT observations convolved with Gaussian ILSs (corresponding to the degraded spectral resolution) for the single-band retrievals. They find that the errors for the low-resolution, single-band retrievals are not terribly worse than those for native GOSAT, as compared to ground-based observations from the TCCON network. They use these results to argue that remote sensing of fossil fuel emissions (such as from power plants) may be possible from low spectral-resolution, single-band sensors with very high spatial resolution (on the order of 50x50 m$^2$).

Overall, the paper is very well written. However, I had some questions about their methodology and conclusions, and recommend publication only after addressing these concerns.

**General Comments**

Because these spectrometers will be for local-scale (power plant, urban scale) domains, the global-scale performance of individual GOSAT 10x10 km^2 really is only a starting point. It would be important to model the potential behavior of such a satellite using an OSSE (Observing System Simulation Experiment) over high-resolution, simulated local-scale domains. The authors should add a (potentially short) discussion of this limitation to the paper.

I have a methodological question as follows. In terms of taking real GOSAT data, and simply convolving it with a wider ILS, it seems like the SNR of the resulting measurement (with 256 channels per band) will be higher than one may actually be able to build in a realistic instrument. For instance, I performed a simulation of simple white noise for ~1300 GOSAT channels spaced every 0.2 cm$^{-1}$ (the approximate channel spacing for GOSAT) between 4740 and 5000 cm$^{-1}$, and had a starting SNR of 700. In the simulation, when I convolved the spectrum (with realistic noise added) with a Gaussian ILS with FWHM=1.3 nm, the resulting SNR was ~ 3400. This was due to the averaging effect of the hi-resolution GOSAT data.

The authors do state (section 2) "Since we want to isolate the effects of spectral resolution and spectral band selection, we do not add extra noise to the convolved spectra." However, they are worried here about the effect of smaller ground pixels. BUT, it seems they are not taking into account this averaging effect "beating down" the native GOSAT noise to unrealistically high SNR values. Here, the final SNR value of 3400 is NOT equal to the GOSAT value of 700, so I think they are not purely

"isolating the effect of spectral resolution" since the SNR values are wildly different. Did the authors examine the resulting SNR of their low-resolution GOSAT measurements, and are they in line with what they would expect from their hypothetical instrument? I realize they somewhat avoid this question by not having a real instrument noise model proposed, but as written, the results may be misleading because they may assume unrealistically high SNR values for any possible instrument. The authors should discuss this point and make it clear. Also, this could be rectified by proposing a realistic instrument noise model, and then ADDING noise to the GOSAT spectrum after convolution with the Gaussian ILS, in order to obtain an SNR in line with a more realistic value.

Another concern is the impact of not using the O2A band. The authors should discuss the feasibility of seeing power plant plumes in the face of realistic pointing errors, and if the pointing will be sufficiently good such that surface pressure estimates from meteorological reanalysis, hypsometrically adjusted to account for the local topography, will be a relatively small error or not.

A critical concern is the ability to properly filter the data. For many XCO2 retrievals, cloud and aerosol filtering is a critical component of any retrieval system, yet this is completely left out of this analysis as the authors start with data pre-filtered using the native GOSAT 3-band retrievals. It is therefore not clear how robust the conclusions would be if the sensor had to solely rely on filtering from a single, low-resolution SWIR band. While this study is a good start, results from a proper simulation-retrieval experiment including the effects of clouds & aerosols and the role of pre-filtering is of critical importance to realistically judge if such a simple sensor could truly determine power plant emissions.

**Specific Comments**

P5L20: You assume 256 spectral channels in a single band. This seems like a high oversampling rate (~3 for both SWIR-1 and SWIR-2), considering that there are roughly 86 fully independent spectral samples in each band, given your proposed resolving powers. This rate appears to have been carefully chosen. Please speak to any knowledge you have on the importance of the spectral oversampling, as it may be an important consideration (for SNR or retrieval accuracy/precision). I just noticed this is also discussed on page 9, but the factor of 3 oversampling is again assumed there, and not questioned or discussed as any kind of instrument parameter to be optimized (in the way that spectral resolution is, in this study).

P6L17. The improvement of your 3-aerosol-parameter retrieval vs. a non-scattering retrieval is curious, consider the extremely low DFS for aerosol you cite (0.38). It therefore seems possible that your results may be sensitive to the prior assumption

on aerosols. How are the aerosol priors for the 3 parameters chosen, and did you test your sensitivity to the aerosol prior, given the low DFS?

Also, is this only for SWIR-2? I would be curious if you attempted scattering retrievals for SWIR-1, to prove that they are no better than non-scattering is right. If my hypothesis is correct, they may be better for the same reason as for SWIR-2 – the the information is more from the prior, and not the measurement itself.

P7L19: The 1.86% scaling factor is interesting. Which way does it go – e.g., do you require a +1.86% scaling of the gas absorption coefficients at 2.01 to match 2.06? Please state this explicitly, as spectroscopists might be interested.

P9: I think it is also important to examine the change in standard deviation (scatter) of GOSAT-TCCON at individual sites, to see if that increases more for some sites over others. The global numbers (3.0 and 3.28 ppm vs. 2.43 ), but it would be interesting to see what these are for individual sites. This information would be usefully presented in a table. In fact, I think a table is important, where the basic information per site is presented (N, mean bias, Stddev). Currently, you try to graphically represent only the per-site bias (in Figure 5).

P9L30: For the parameter correlations, I think you should also look at the retrieved aerosol parameters from SWIR-2 when looking at the XCO2 from SWIR-2. At least check it. I would be surprised if those correlations were not higher than they are for the parameters from the native retrieval, which is VERY different (3 bands, high spectral resolution, etc).

Section 4: You should state the purpose of the extensive comparison of the modified SWIR-1 and SWIR-2 retrievals to the native GOSAT retrievals. You take the native GOSAT retrievals as the reference, but they are NOT truth. So the value of several of the Figures (7-11) is dubious. You could shorten the paper by removing some of these figures, since you honestly do not know, in many instances, whether the low-resolution, single band retrievals are actually less accurate than the high-resolution, 3-band retrievals.

P11/Fig 7: What are the R (or R^2) values for SWIR-1 and SWIR-2 vs. Native? These are useful to see as well. I suggest also including these numbers in Fig. 9, and perhaps the corresponding main text as well. Ie, is 90% of the variance explained, or 50%? Etc.

P17/Fig 12: Per the discussion of the SNR, this relates to my general comment above, about whether the SNRs you actually ran tests on are even remotely achievable. In practice, most instrument builders will tell you that there is a trade off between SNR and spectral resolution. They are not independent, as this work seems to imply. This should be stated more clearly. As I said above, my preference would be to consult with instrument builders and find out what are reasonable noise models for

the type of instrument you want to build, and actually run retrieval tests on those, rather than on the likely unrealistic SNR values within this work.

---

## Author Response (AR1)

**Manuscript: "Spectral Sizing of a Coarse Spectral Resolution Satellite Sensor for XCO2" by JS Wilzewski et al.**

**Reply to interactive comment by anonymous reviewer #1**

We thank the reviewer for the helpful comments to our manuscript. Below we repeat the reviewer's questions in **bold** font and subsequently provide our responses.

1. It is unclear to me how column mean dry air mole fractions of CO2 are obtained in a retrieval without NIR band, i.e. in a retrieval where no O2 column is estimated. Where is the information on O2 taken from? From surface pressures from a weather prediction model? How does that add to the overall uncertainty? Isn't the retrieval very sensitive to topographic variations and thus to the pointing accuracy of the instrument in this case?

Column averaged dry air mole fractions of  $CO_2$ ,  $XCO_2$ , are calculated by deviding the retrieved CO2 concentrations by the airmass below the satellite. In our work, the airmass is determined from a global digital elevation model (NASA's Shuttle Radar Tomography Mission – SRTM) together with surface pressure reanalyses from ECMWF (ERA Interim). This is the standard way how our native RemoTeC algorithm works for trace gas retrievals from the GOSAT, OCO-2 and TROPOMI satellite instruments.

The point raised in your comment about pointing accuracy is very important. Naturally, if our proposed instrument was pointed towards a target site on a terrain with a great slope, pointing errors would be translated into elevation errors/XCO2 errors (20 m of elevation error would result in roughly 1 ppm of XCO2 error). Thus, errors in the calculation of the airmass are part of the overall uncertainty found in our analysis. But, these error contributions are equal for the native and reduced-resolution retrievals because the calculation of airmass is the same. In fact, not shown in the paper, we tried to refine our analysis by looking at localized signals above the urban area of Los Angeles. There, we found that uncertainties in GOSAT's pointing can induce significant errors relatated to the airmass calculation.

To clarify how airmass is obtained, we added the following on page 7, line 23 – 28: "For both, native GOSAT and degraded SWIR configurations, airmass information is derived from ECMWF surface pressure reanalyses (ERA-Interim) and topographic data from the Shuttle Radar Tomography Mission (SRTM). For each sounding, we use ECMWF and SRTM data to calculate the ground-pixel average surface pressure and the corresponding dry airmass. This is the standard operation procedure for RemoTeC trace gas retrievals from the GOSAT, OCO-2 and TROPOMI satellite instruments. Errors in the calculation of the airmass can be caused by erroneous satellite pointing; these errors are part of the overall errors reported for the TCCON validation sites (section 3)."

2. A problem not really addressed in the study is the fact that coarser spectral resolution instruments tend to have larger uncertainties in the spectral calibration. The retrieval can account for spectral shifts, but this is more difficult in case of coarsely resolved spectra. What were the assumptions regarding spectral calibration uncertainties and how would that affect the conclusions?

There were no assumptions regarding spectral calibration uncertainties in this study. Spectral shifts are free parameters in our retrieval for both, the native and the reduced resolution setups. We start with the standard spectral calibration provided in the GOSAT L1B data files. Then, the retrievals shift the simulated observations to minimize the least-squares difference to the observations. Any errors caused by interferences of adjusting the spectral shifts and fitting  $XCO_2$  are contributors to the errors that we discuss in the paper. However, we have no indication (e.g. particularly large uncertainties of the spectral shift parameters) that spectral shifting is a large error contribution.

3. Only quality-screened cloud-free GOSAT spectra were used in the analysis. How much does that screening depend on the information in the NIR and SWIR channels? Or in other words, how much more difficult would quality/cloud screening be for an instrument with a single SWIR channel? This seems important to me, since only a small proportion of pixels usually survive the strict quality flagging required for satellite CO2 retrievals.

The question of quality screening has not been addressed in this study. Unfortunately, we cannot afford the computational costs to reprocess the entire GOSAT dataset (including all the cloudy data) of the years 2009 to 2016 that we used in the study. Thus, we cannot give a quantitative reply to this remark, but we argue that while cloud detection would certainly be more challenging with a coarse resolution 1-Band configuration, the SWIR-2 spectral range offers the possibility to construct a decent cloud filter. To this end, one could make use of the two  $CO_2$  bands in this window, which have the advantage of having very different optical depths.  $XCO_2$  retrievals from either band should then be consistent in cloud-free scenes and different in complicated scenes. Yet, to assess this screening procedure, a follow up study is necessary.

We now mention in the paper that we did not carry out a cloud filtering exercise on page 5, line 8: "Due to computational costs, we restrict our analysis to cloud-free, quality screened soundings over land as identified by the native GOSAT retrievals of the RemoTeC algorithm...".

Also, we added a discussion of the SWIR cloud filter option in the discussion (page 18, line 4 - 8): "Additionally, the SWIR-2 seems better suited for the construction of a cloud filter, because its CO2 bands have very different optical depths. Similar to the cloud filter currently in use for GOSAT measurements, one could retrieve XCO2 from the two SWIR-2 bands individually and filter for discrepancies. This scheme should be tested in the future."

Since the main application of the sensor will be point-source detection and quantification, a future study should focus on local rather than global scales as done here. The recent study of Cusworth et al. (2019; https://doi.org/10.5194/amt-2019-202), for example, shows that local plume detection can be significantly affected by retrieval errors which are correlated with surface reflectance. The spectral resolution of the instrument proposed here may be high enough to mitigate such problems, but this aspect should receive more attention in a future study.

Thank you for pointing out the study by Cusworth et al., which we now cite in the discussion (page 17, line 30): "Surface reflectance has been shown to be a central driver in methane retrieval precision by Cusworth et al. (2019)". We also agree that local scale phenomena related to surface reflectance and plume detection need to be investigated further in coming studies. Manuscript: "Spectral Sizing of a Coarse Spectral Resolution Satellite Sensor for XCO2" by JS Wilzewski et al.

**Reply to interactive comment by anonymous reviewer #2**

We thank the reviewer for the helpful comments to our manuscript. Below we repeat the reviewer's questions in **bold** font and subsequently provide our responses.

**General Comments**

**The authors traded-off the spectral resolution but performance with and without O2A is not clear. Did they consider an O2A spectrometer with moderate spectral resolution?**

We realize that knowledge of particle scattering in the atmosphere would be improved by observing the oxygen A-Band. Any information, even at coarse spectral resolution, on  $O_2$  absorption in the NIR would be useful to characterize aerosol properties. However, we have not considered an additional spectrometer as this would significantly increase cost and mass of the proposed instrument that we envision to be employed in fleets of relatively inexpensive and small satellites. In terms of possibly losing performance due to uncertainties in the airmass, because of our one band set-up, we would like to emphasize that RemoTeC does not rely on the  $O_2$  A-band to calculate airmass. Instead, information from prior topography and meteorology (such as surface pressure values from the ERA Interim product by ECMWF) is used to determine the air mass below the satellite.

A short discussion of how RemoTeC calculates airmass was added on page 7, line 23 – 27: "For both, native GOSAT and degraded SWIR configurations, airmass information is derived from ECMWF surface pressure reanalyses (ERA-Interim) and topographic data from the Shuttle Radar Tomography Mission (SRTM). For each sounding, we use ECMWF and SRTM data to calculate the ground-pixel average surface pressure and the corresponding dry airmass. This is the standard operation procedure for RemoTeC trace gas retrievals from the GOSAT, OCO-2 and TROPOMI satellite instruments."

**GOSAT has measured several data over cities such as Tokyo and LA using its target observation function. The authors can pick up and discuss aerosol effect over cities.**

We investigated the possibility to focus on localized signals by looking at

target observations of the Los Angeles basin. However, the data were still too sparse to evaluate the effect of aerosols in a significant matter.

**They also concluded that dust is the largest error source by specifying latitudinal ranges. GOSAT has observed desert area such as Sahara and Arabian Desert with its medium gain. The authors can analyze directly by picking up medium gain data**

Our analysis shows that errors correlate with the desert latitudes. This is apparent from our analysis, which is based on a mix of high and medium gain spectra (medium gain measurements account for ca. 12 % of the dataset). We do not see what additional information an analysis of medium gain data alone would provide.

**Specific Comments**

(1) Page 1, Abstract The spectral resolution coarser than native GOSAT and the single-band of CO2 without O2A band are both key parts of this study. However, the latter is not clearly mentioned in the abstract.

We have emphasized that we carry out single-band retrievals in the abstract (page 1, line 11): "...and we evaluate single-band retrievals...".

**(2) page 6, Line 18, "non-scattering retrieval", Page 8, Line 9, "the non-scattering SWIR-1 retrieval" Brief description is needed.**

Non-scattering retrievals refer to retrievals where scattering by particles is neglected.

We added an explanation (page 6, line 8 - 9): "This approach, which is essentially a transmittance calculation along the geometric lightpath, is hereafter referred to as non-scattering retrieval."

**(3) Page 6, Line 20, "More than 75 % of all retrievals converge at any given FWHM that we consider in this study." It is difficult to understand**

This statement refers to the fact that the retrieval algorithm converges towards a solution after a reasonable number of iterations for the majority of retrievals we perform. As the degree of freedom for signal for the aerosol parameter retrieval varies with resolving power, the retrieval becomes more or less tightly regularized so that it may not find the global minimum of the cost function. We added to a sentence in the paper to make this clearer (page 7, line 9 - 10): "Although variations in DFS may lead to changes in the ability of the retrieval algorithm to converge towards the minimum of the cost function, ...".

**(4) Page 7, Line 19, "1.856%" It is not clear. Is it 1.8% of XCO2? 7.4 ppm?**

To make the spectroscopic cross sections of  $CO_2$  near 2  $\mu$ m consistent, we apply a scaling factor to the strong  $CO_2$  band. This factor was determined from calculating XCO2 from the two bands separately. The referee is right that it was not clear which way our scaling of the XCO2 cross sections at 2.01  $\mu$ m goes. We added the information in the text (page 8, line 1): "i.e. the cross sections of the 2.01  $\mu$ m band need to be scaled by 0.981".

**(5) Page 10, Figure 4 Use of three individual figures will become clearer.**

We have updated the figure accordingly.

**(6) Page 16, Line 30 "an additional aerosol sensor may help" The largest error source seems to be vertical profile of particles. Conventional aerosol imager provides horizontal distribution only. Which kind of sensor do authors consider?**

Ideally, we want an aerosol instrument that provides multi-angle, radiance and polarization information over a wide spectral range (from the UV to the NIR) while the instrument is sufficiently compact to fit on a small satellite. Hasekamp et al. (2019) describe such an instrument to be deployed on the NASA PACE mission (we included a reference to this work in the manuscript on page 17, line 17).

**Technical Corrections (1) Page 6, Line 11 XH2O Definition of 'XH2O' should be described**

A short description was added in the text (page 5, line 27 - 28): "throughout this work X*molecule* refers to the column-averaged dry air mole fraction of a molecule"

(2) Page 12, Figure 12 At present, it is monochromatic. It should be a color figure such as figures 6 and 11. Grey line is difficult to

**see.**

This Figure was updated with the same color map as Figures 6 and 11.

**References**

Hasekamp, O. P., Fu, G., Rusli, S. P., Wu, L., Di Noia, A., aan de Brugh, J., Landgraf J., Smit, J. M., Rietjens, J., van Amerongen, A.: Aerosol measurements by SPEXone on the NASA PACE mission: expected retrieval capabilities, Journal of Quantitative Spectroscopy and Radiative Transfer, 227, 170 – 184, https://doi.org/10.1016/j.jqsrt.2019.02.006

**Manuscript: "Spectral Sizing of a Coarse Spectral Resolution Satellite Sensor for XCO2" by JS Wilzewski et al.**

**Reply to interactive comment by anonymous reviewer #3**

We thank the reviewer for the helpful comments to our manuscript. Below we repeat the reviewer's questions in **bold** font and subsequently provide our responses.

**General Comments**

Because these spectrometers will be for local-scale (power plant, urban scale) domains, the global-scale performance of individual GOSAT 10x10 km2 really is only a starting point. It would be important to model the potential behavior of such a satellite using an OSSE (Observing System Simulation Experiment) over high-resolution, simulated local-scale domains. The authors should add a (potentially short) discussion of this limitation to the paper.

We agree that most of the analyses performed in this work are just a starting point towards evaluating a possible future  $CO_2$  sensor. It will certainly be crucial to carry out detailed simulations of this proposed spectrometer with a thorough discussion of the actual instrument design and noise performance for representative local-scale domains. In the present manuscript we focus on investigating whether a coarse resolution, single band observation configuration could generally deliver sufficient information such that a meaningful retrieval of  $XCO_2$  can be made. Details of the satellite sensor shall be studied in a forthcoming study, currently under preparation in our group.

We added "A forthcoming study addressing these aspects of the proposed sensor is currently under preparation." in the manuscript on page 18, line 8 - 9.

I have a methodological question as follows. In terms of taking real GOSAT data, and simply convolving it with a wider ILS, it seems like the SNR of the resulting measurement (with 256 channels per band) will be higher than one may actually be able to build in a realistic instrument. For instance, I performed a simulation of simple white noise for 1300 GOSAT channels spaced every 0.2 cm-1 (the approximate channel spacing for GOSAT) between 4740 and 5000 cm-1, and had a starting SNR of 700. In the simulation, when I convolved the spectrum (with realistic noise added) with a Gaussian ILS with FWHM=1.3 nm, the resulting SNR was  $\sim$ 3400. This was due to the averaging effect of the

hi-resolution GOSAT data.

The authors do state (section 2) "Since we want to isolate the effects of spectral resolution and spectral band selection, we do not add extra noise to the convolved spectra." However, they are worried here about the effect of smaller ground pixels. BUT, it seems they are not taking into account this averaging effect 'beating down' the native GOSAT noise to unrealistically high SNR values. Here, the final SNR value of 3400 is NOT equal to the GOSAT value of 700, so I think they are not purely "isolating the effect of spectral resolution" since the SNR values are wildly different. Did the authors examine the resulting SNR of their low-resolution GOSAT measurements, and are they in line with what they would expect from their hypothetical instrument? I realize they somewhat avoid this question by not having a real instrument noise model proposed, but as written, the results may be misleading because they may assume unrealistically high SNR values for any possible instrument. The authors should discuss this point and make it clear. Also, this could be rectified by proposing a realistic instrument noise model, and then ADDING noise to the GOSAT spectrum after convolution with th Gaussian ILS, in order to obtain an SNR in line with a more realistic value.

It is evident that the effect of convolving the native GOSAT spectra with a wider ILS (sampled by 3 detector pixels) results in higher SNR per pixel for the setup with reduced spectral resolution then for the native configuration. And, indeed, we do not compensate for the "beating down" of the noise by adding extra noise (as mentioned by the manuscript).

Adding noise to the spectra would introduce an additional artificial element (besides the coarse ILS) to our analysis. We want to stick as close as possible to real measurements and, as stated by the manuscript, we want to isolate the one effect (i.e. coarse ILS).

Errors in native GOSAT retrievals are not dominated by noise. In fact, an SNR of 700 (at the radiance continuum?) as assumed by the reviewer is by far better than the observed spectral fitting residuals which are for the most part dominated by systematic patterns (unresolved scattering effects, spectroscopic errors, unaccounted instrument characteristics). Likewise, the noise errors on retrieved XCO2 are typically a factor 2-4 smaller than the standard deviations found when comparing to validation data (see also our figure 12). Thus, for a GOSAT-like setup, noise is a minor contributor to the errors. The noise for convolved and unconvolved spectra might be "wildly" different, but, for both, it is small compared to other sources of error. Accepting that the noise is small makes it straightforward

to evaluate these other sources of error e.g. through the parameter correlations shown in the paper, which we chose to be the focus of the present paper.

Adding noise to the spectra to mimic a new sensor with fine ground resolution would result in a different paper. We would need to discuss the instrument optical and electronic setup and describe the noise model. Such a paper is in preparation including a noise evaluation with simulated data. The present paper aims at discussing whether it is reasonable at all to try out a coarse-spectral-resolution configuration.

Essentially, our results are representative under circumstances where the noise can be assumed small compared to other sources of error. The next paper will address how to build the instrument, for what scenes noise is indeed negligible, and what to expect if noise becomes large for dark surfaces. To make these aspects clear, we add the following paragraph to the manuscript:

"Our approach essentially relates to conditions under which the detector noise is negligible as typical for GOSAT. Under such conditions, other sources of error can be addressed e.g. through evaluating geophysical parameter correlations (section 3 and 4). A forthcoming study will discuss noise performance and retrieval simulations for a hypothetical instrument design." (page 5, line 20 - 23).

Another concern is the impact of not using the O2A band. The authors should discuss the feasibility of seeing power plant plumes in the face of realistic pointing errors, and if the pointing will be sufficiently good such that surface pressure estimates from meteorological reanalysis, hypsometrically adjusted to account for the local topography, will be a relatively small error or not.

As the proposed sensor will have imaging ability, the spectrometer shows promise to have a good pointing knowledge. Any errors in pointing may be 'recalibrated' when scenes with prominent surface reflectance features, such as shorelines, etc., are observed. We expect that even if pointing accuracy is low, one would be able to obtain a good correction in order to correctly calculate airmass for the  $XCO_2$ retrieval.

We added "Errors in the calculation of the airmass can be caused by erroneous satellite pointing; these errors are part of the overall errors reported for the TC-CON validation sites (section 3)." (page 7, line 26 - 27).

A critical concern is the ability to properly filter the data. For many XCO2 retrievals, cloud and aerosol filtering is a critical component of any retrieval system, yet this is completely left out of this analysis as the authors start with data pre-filtered using the native GOSAT 3-

band retrievals. It is therefore not clear how robust the conclusions would be if the sensor had to solely rely on filtering from a single, lowresolution SWIR band. While this study is a good start, results from a proper simulation-retrieval experiment including the effects of clouds & aerosols and the role of pre-filtering is of critical importance to realistically judge if such a simple sensor could truly determine power plant emissions.

We would have liked to analyze the impact on cloud-screening, however, due to computational costs, we could not. It should be pointed out that the SWIR-2 configuration, which is favored for the future instrument, has two  $CO_2$  absorption bands with very different optical depths, which opens up an avenue to set-up a cloud filter using the SWIR-2 window alone. By retrieving  $XCO_2$  from both  $CO_2$  bands, one could filter for large discrepancies caused by the presence of clouds. This is a variant of the cloud filter currently used for the native GOSAT soundings. The actual implementation and verification of this approach must be postponed to a future study.

We now mention in the paper that we did not carry out a cloud filtering exercise on page 5, line 8: "Due to computational costs, we restrict our analysis to cloudfree, quality screened soundings over land as identified by the native GOSAT retrievals of the RemoTeC algorithm...".

Also, we added a discussion of the SWIR cloud filter option in the discussion (page 18, line 4-7): "Additionally, the SWIR-2 seems better suited for the construction of a cloud filter, because its CO2 bands have very different optical depths. Similar to the cloud filter currently in use for GOSAT measurements, one could retrieve XCO2 from the two SWIR-2 bands individually and filter for discrepancies. This scheme should be tested in the future."

**Specific Comments**

P5L20: You assume 256 spectral channels in a single band. This seems like a high oversampling rate ( $\sim$ 3 for both SWIR-1 and SWIR-2), considering that there are roughly 86 fully independent spectral samples in each band, given your proposed resolving powers. This rate appears to have been carefully chosen. Please speak to any knowledge you have on the importance of the spectral oversampling, as it may be an important consideration (for SNR or retrieval accuracy/precision). I just noticed this is also discussed on page 9, but the factor of 3 oversampling is again assumed there, and not questioned or discussed as any kind of instrument parameter to be optimized (in the way that spectral resolution is, in this study).

We have assumed a spectral sampling ratio of three throughout this work. A sampling ratio of 2 would be the lower limit according to Nyquist's theorem. Generally, the higher the sampling ratio, the better. Detectors with a very high number of pixels (e.g 2000 pixels) could enable a significantly higher sampling ratio. Yet, previous space-based CO2 missions have been successful by spectrally over-sampling the FWHM by a factor 2-3 (e.g. GOSAT, OCO-2, OCO-3, TanSat). Thus our choice of sampling ratio is based on what is currently in use for similar sensors.

**P6L17. The improvement of your 3-aerosol-parameter retrieval vs. a non-scattering retrieval is curious, consider the extremely low DFS for aerosol you cite (0.38). It therefore seems possible that your results may be sensitive to the prior assumption on aerosols. How are the aerosol priors for the 3 parameters chosen, and did you test your sensitivity to the aerosol prior, given the low DFS?**

Given that the retrievals estimate 3 aerosol parameters with little DFS, the retrievals, by definition of DFS, depend on the a priori. We have conducted a sensitivity study how various aerosol priors map into XCO2 errors. As prior aerosol we had selected reasonable numbers for scattering optical depth ( $\tau$ =0.1), scattering layer height ( $z_{par}$ =3000 m) and size parameter ( $\alpha_{par}$ =3.5) throughout the study. These values are routinely used as prior for GOSAT retrievals with RemoTeC. Table 1 shows the changes in scatter around TCCON as well as the changes in correlation coefficients for SWIR-2 retrievals at 1.29 nm resolution with changed aerosol priors. As we only have ~0.4 degrees of freedom to be distributed to the fit of three aerosol parameters, it is clear that the aerosol prior can have an impact on retrieval performance.

We find that our results are moderately sensitive to small changes in  $\tau$  or  $z_{par}$ , while larger variations in the prior have a big impact on XCO2 retrieval performance. For instance, changing  $\tau$  by a factor 2 or  $\frac{1}{2}$  leads to relatively small deviations from our benchmark SWIR-2 retrieval regarding scatter around TCCON and geophysical correlations on a global scale. We observe  $\sigma_{TCCON} = 3.19$  ppm for  $\tau=0.05$  and  $\sigma_{TCCON} = 3.50$  ppm for  $\tau=0.2$ . Correlation coefficients to albedo and other geophysical parameters (as in Fig. 11 of the manuscript) are collected in Table 1. Changes in retrieval performance also occur for small changes in the initial scattering layer height. For prior layer heights of 1000 m and 5000 m, the standard deviation of SWIR-2 retrievals around colocated TCCON data amounts to 3.32 ppm and 3.71 ppm, respectively. This indicates a stronger dependence on scattering layer height priors than on optical depth priors. A significant change in retrieval performance occurs for a prior aerosol scenario, where rather large scattering particles are placed at the top of the troposphere ( $\tau=0.07 \text{ } z_{par}=11600 \text{ } \text{m} \alpha_{par}=3.67$ ). In this case,  $\sigma_{TCCON}=4.14$  ppm is higher than for all other aerosol prior options we studied here.

As a result, extreme prior aerosol values have to be avoided for our retrievals. This sensitivity study shows that the retrieval performance of the proposed sensor may be enhanced by a few tenths of a ppm by using a good aerosol prior. An additional aerosol sensor would help to inform and optimize the retrievals.

We added "An investigation of the impact of the aerosol priors on retrieval performance showed that SWIR-2 XCO2 is only moderately sensitive to the aerosol priors. For instance, varying aerosol prior optical depth by a factor of two or one half resulted in small changes in standard deviations around TCCON (+0.22 ppm and -0.08 ppm, respectively). Changing scattering layer height priors to  $z_{par}=1000$  m or  $z_{par}=5000$  m increased scatter around TCCON by +0.04 ppm and +0.43 ppm, respectively. Similarly, scatter around TCCON changes by +0.22 ppm and -0.05 ppm if  $\alpha_{par}$  is set to 3.0 and 5.0, respectively." in the manuscript (page 10, line 20 – page 11, line 2).

| Aerosol prior                                                         | $\sigma_{TCCON}$ / ppm | R(albedo) | R(SOT) | $R(N_{par})$ | $R(z_{par})$ | $R(\alpha_{par})$ |
|-----------------------------------------------------------------------|------------------------|-----------|--------|--------------|--------------|-------------------|
| $\tau = 0.1$ $z_{par} = 3000 \text{ m}$ $\alpha_{par} = 3.5$          | 3.28                   | -0.18     | 0.26   | -0.5         | 0.21         | 0.01              |
| $\tau = 0.07$
$z_{par} = 11600 \text{ m}$
$\alpha_{par} = 3.67$ | 4.14                   | -0.48     | 0.17   | -0.5         | 0.17         | 0.09              |
| $\tau = 0.05$
$z_{par} = 3000 \text{ m}$
$\alpha_{par} = 3.5$   | 3.19                   | 0.04      | 0.31   | -0.45        | 0.21         | -0.07             |
| $\tau = 0.2$
$z_{par} = 3000 \text{ m}$
$\alpha_{par} = 3.5$    | 3.50                   | -0.30     | 0.21   | -0.50        | 0.19         | 0.07              |
| $\tau = 0.1$
$z_{par} = 1000 \text{ m}$
$\alpha_{par} = 3.5$    | 3.32                   | 0.20      | 0.31   | -0.4         | 0.19         | -0.11             |
| $\tau = 0.1$
$z_{par} = 5000 \text{ m}$
$\alpha_{par} = 3.5$    | 3.71                   | -0.36     | 0.21   | -0.47        | 0.20         | 0.06              |
| $\tau = 0.1$
$z_{par} = 3000 \text{ m}$
$\alpha_{par} = 3.0$    | 3.42                   | -0.24     | 0.23   | -0.49        | 0.20         | 0.07              |
| $\tau = 0.1$
$z_{par} = 3000 \text{ m}$
$\alpha_{par} = 5.0$    | 3.23                   | -0.05     | 0.3    | -0.5         | 0.2          | -0.09             |

Table 1: Comparison of the effect of different aerosol priors on standard deviation of retrieval results around TCCON (" $\sigma_{TCCON}$ ") and on the correlation coefficients ("R(X)") with respect to geophysical parameters (albedo at 2.1 nm, SOT, particle amount, scattering layer height and size parameter) as in Fig. 11 of the manuscript. The highlighted row shows the parameters for the prior with which we have carried out the calculations for the manuscript. The respective aerosol prior is shown in the first column.

Also, is this only for SWIR-2? I would be curious if you attempted scattering retrievals for SWIR-1, to prove that they are no better than non-scattering is right. If my hypothesis is correct, they may be better for the same reason as for SWIR-2? the the information is more from the prior, and not the measurement itself.

We did attempt to include scattering in the SWIR-1 retrievals as mentioned on page 7, line 1-2, but even at native GOSAT spectral resolution, a SWIR-1 single band retrieval accounting for scattering typically has an average of 0.24 degrees of freedom for three aerosol parameters. At coarse spectral resolution we encountered low information content and worse retrieval performance with respect to scatter around TCCON. Thus, neglecting aerosol particles in the retrievals seemed the better choice. We added "the SWIR-1 band suffers from low information content and results in worse XCO2 retrieval performance than under the non-scattering assumption" on page 7, line 2-3 in the manuscript.

P7L19: The 1.86% scaling factor is interesting. Which way does it go? e.g., do you require a +1.86% scaling of the gas absorption coefficients at 2.01 to match 2.06? Please state this explicitly, as spectroscopists might be interested.

This was indeed unclear. We have added "(i.e. cross sections of the 2.01  $\mu$ m band need be scaled by 0.981)" in the manuscript (page 7, line 35) to explain this scaling.

P9: I think it is also important to examine the change in standard deviation (scatter) of GOSAT-TCCON at individual sites, to see if that increases more for some sites over others. The global numbers (3.0 and 3.28 ppm vs. 2.43), but it would be interesting to see what these are for individual sites. This information would be usefully presented in a table. In fact, I think a table is important, where the basic information per site is presented (N, mean bias, Stddev). Currently, you try to graphically represent only the per-site bias (in Figure 5).

This information is indeed useful and we have decided to expand Fig. 5 to also show scatter around TCCON at individual sites (we also changed the caption accordingly). Furthermore, the figure was updated to contain information about the number of colocated soundings at each station. In addition, we added Table 2 in the form of a supplementary material.

Figure 1: Comparison of retrieval performances at individual TCCON stations sorted north to south. Marker size indicates amount of colocated soundings at each station. Left: Station-by-station mean differences between TCCON and the native (black), SWIR-1 (red), and SWIR-2 (blue) retrievals from GOSAT. The standard deviation of mean differences among the stations,  $\sigma$ , amounts to 0.94 ppm (native), 0.99 ppm (SWIR-1) and 0.97 ppm (SWIR-2). Right: Scatter around TCCON per station for the native, SWIR-1, and SWIR-2 retrievals. Vertical lines mark the average standard deviations (native: 2.43 ppm, SWIR-1: 3.00 ppm, SWIR-2: 3.28 ppm).

We added "Figure 5 also shows  $XCO_2$  retrieval standard deviations per TC-CON station. The corresponding data for retrieval performance at individual sites can be found in the supplementary materials." on page 10, line 1 - 3.

| TCCON site     | N    |        |        | Bias / ppm |        |        | $\sigma / \text{ppm}$ |        |        |
|----------------|------|--------|--------|------------|--------|--------|-----------------------|--------|--------|
|                | FP   | SWIR-1 | SWIR-2 | FP         | SWIR-1 | SWIR-2 | FP                    | SWIR-1 | SWIR-2 |
| Sodankyla      | 217  | 211    | 217    | -0.38      | -0.83  | 2.55   | 2.08                  | 2.71   | 3.47   |
| Bialystok      | 714  | 673    | 708    | -0.76      | -0.88  | -0.6   | 2.14                  | 2.82   | 3.46   |
| Bremen         | 229  | 218    | 229    | -0.28      | -0.75  | 1.49   | 2.43                  | 3.11   | 3.47   |
| Karlsruhe      | 512  | 478    | 512    | -0.82      | -0.66  | 0.39   | 2.49                  | 3.26   | 3.86   |
| Paris01        | 215  | 211    | 214    | -1.52      | -1.37  | -0.52  | 2.61                  | 3.4    | 3.2    |
| Orleans        | 740  | 712    | 736    | -0.77      | -1.02  | 0.28   | 2.06                  | 3.13   | 3.52   |
| Garmisch       | 493  | 462    | 493    | -0.4       | -0.47  | 0.39   | 2.14                  | 2.93   | 3.7    |
| Zugspitze      | 69   | 66     | 69     | -1.47      | -1.24  | 0.83   | 2.85                  | 3.04   | 4.24   |
| Park Falls     | 940  | 905    | 896    | -0.53      | -0.35  | -0.21  | 2.09                  | 2.84   | 3.41   |
| Rikubetsu      | 68   | 60     | 68     | -1.47      | -1.52  | -1.02  | 1.82                  | 2.87   | 3.12   |
| Indianapolis01 | 195  | 193    | 188    | 0.18       | -0.15  | 1.24   | 1.84                  | 2.62   | 2.99   |
| 4Corners       | 45   | 30     | 34     | -0.6       | 0.14   | 0.29   | 3.36                  | 2.24   | 2.19   |
| Lamont         | 5047 | 4939   | 4208   | -0.62      | -0.02  | -1.15  | 1.98                  | 2.83   | 2.62   |
| Anmeyondo      | 9    | 9      | 9      | -1.1       | 0.53   | 1.05   | 2.75                  | 2.29   | 2.63   |
| Tsukuba        | 837  | 731    | 830    | 1.15       | 0.16   | 0.63   | 2.81                  | 3.38   | 3.83   |
| Edwards        | 1666 | 1575   | 1462   | 1.98       | 2.07   | 2.31   | 2.64                  | 2.95   | 3.0    |
| JPL02          | 713  | 652    | 659    | 0.52       | 0.15   | 0.36   | 1.95                  | 2.77   | 2.79   |
| Pasadena01     | 2209 | 2084   | 1979   | 0.27       | 0.09   | 0.7    | 2.58                  | 2.97   | 3.05   |
| Saga           | 293  | 264    | 287    | -0.2       | -1.21  | -0.96  | 2.29                  | 3.31   | 3.53   |
| Hefei          | 159  | 148    | 159    | -0.96      | -0.09  | -0.77  | 2.24                  | 3.2    | 3.0    |
| Darwin         | 1521 | 1510   | 1404   | 1.07       | 0.63   | -0.51  | 1.59                  | 2.14   | 2.43   |
| Wollongong     | 1029 | 975    | 974    | -0.4       | -1.05  | -0.58  | 2.22                  | 2.7    | 3.12   |
| Lauder02       | 65   | 61     | 65     | -2.22      | -1.92  | -0.06  | 2.04                  | 2.61   | 3.48   |
| Lauder01       | 17   | 15     | 17     | -1.48      | -3.11  | -0.25  | 2.7                   | 2.67   | 4.1    |

Table 2: Comparison of full-physics (FP), SWIR-1 and SWIR-2 retrievals for soundings colocated with individual TCCON stations. Stations are sorted north to south in the first column. Number of soundings (second column from left), mean differences between the present retrievals and TCCON ("bias"; third column) and standard deviation ("scatter"; last column).

P9L30: For the parameter correlations, I think you should also look at the retrieved aerosol parameters from SWIR-2 when looking at the XCO2 from SWIR-2. At least check it. I would be surprised if those correlations were not higher than they are for the parameters from the native retrieval, which is VERY different (3 bands, high spectral resolution, etc). We analyzed correlations of  $\Delta XCO_2$  (SWIR-2 - TCCON) with aerosol parameters retrieved from the SWIR-2 configuration. We find that, in comparison to the correlations to the full physics aerosol parameters we used previously,

- correlation with  $N_{par}$  changes from -0.05 (FP aerosol parameters) to -0.21 (SWIR-2 aerosol parameters)
- correlation with  $z_{par}$  changes from -0.32 (FP aerosol parameters) to 0.05 (SWIR-2 aerosol parameters)
- correlation with  $\alpha_{par}$  changes from 0.08 (FP aerosol parameters) to 0.29 (SWIR-2 aerosol parameters)

As the reviewer argued, it does make a difference which aerosol parameters are used here. Interestingly, the SWIR-2  $XCO_2$  error with respect to TCCON correlates more strongly with particle amount and size in case of the SWIR-2 aerosol retrieval than for the FP aerosol parameters. Scattering layer height, however, correlates less with retrieval errors when the SWIR-2 layer height is used.

Section 4: You should state the purpose of the extensive comparison of the modified SWIR-1 and SWIR-2 retrievals to the native GOSAT retrievals. You take the native GOSAT retrievals as the reference, but they are NOT truth. So the value of several of the Figures (7-11) is dubious. You could shorten the paper by removing some of these figures, since you honestly do not know, in many instances, whether the low-resolution, single band retrievals are actually less accurate than the high-resolution, 3-band retrievals.

The native GOSAT retrievals were shown to compare better to TCCON than the coarse resolution SWIR retrievals in section 3. Of course, the native-GOSAT XCO2 data are not perfect, but at least they have been shown to be useful in many studies of GOSAT measurements. For this reason, we illustrate retrieval errors with respect to the native GOSAT retrieval (e.g. Figs 7,9,10,11). We do believe it is helpful to show these plots as they give insight into SWIR retrieval errors caused by geophysical dependencies on a global scale. A comparison to TCCON is limited to the site locations of the network and does not reflect variations in geophysical parameters that are observed globally. These plots also help to demonstrate limitations of the proposed sensor. At the same time they help to make the point that our coarse resolution approach is generally comparable to the native RemoTeC GOSAT XCO2 product.

P11/Fig 7: What are the R (or  $R^2$ ) values for SWIR-1 and SWIR-2

vs. Native? These are useful to see as well. I suggest also including these numbers in Fig. 9, and perhaps the corresponding main text as well. Ie, is 90% of the variance explained, or 50%? Etc.

We have included Pearson's correlation coefficient in the plots. For both, SWIR-1 and SWIR-2, the value is 0.90.

P17/Fig 12: Per the discussion of the SNR, this relates to my general comment above, about whether the SNRs you actually ran tests on are even remotely achievable. In practice, most instrument builders will tell you that there is a trade off between SNR and spectral resolution. They are not independent, as this work seems to imply. This should be stated more clearly. As I said above, my preference would be to consult with instrument builders and find out what are reasonable noise models for the type of instrument you want to build, and actually run retrieval tests on those, rather than on the likely unrealistic SNR values within this work.

As we discuss in the introduction of the paper, several authors have proposed pursuing a coarse spectral resolution spectrometer for the detection of localized  $CO_2$  and  $CH_4$  emissions from space (e.g. Dennison et al. (2013), Thorpe et al. (2016)). In light of these previous studies, we investigate here whether a  $CO_2$  satellite monitoring mission would be generally within the realms of possibility and which spectral resolutions are favorable. Instrument design is currently in progress and will be the subject of a forthcoming paper. From a technical point of view, the instrument will require a large telescope (e.g. 15 cm diameter) and a fast optics (f-number < 2.5).

Figure 4:

updated Figure shows FP, SWIR-1 and SWIR-2 retrieval standard deviations with respect to TCCON side-by-side.

Page 10, line 2 - 4:

added "Figure 5 also shows XCO2 retrieval standard deviations per TCCON station. The corresponding data for retrieval performance at individual sites can be found in the supplementary materials."

Figure 5:

Added three more panels to the plot showing scatter around TCCON. Marker size now reflects number of individual soundings available at each TCCON site.

Page 10, line 19: added "( $\tau$ =0.1, zpar=3000 m,  $\alpha_{par}$ =3.5)"

Page 10, line 21 - page 11, line 3:

added "An investigation of the impact of the aerosol priors on retrieval performance showed that SWIR-2 XCO2 is only moderately sensitive to the aerosol priors. For instance, varying aerosol prior optical depth by a factor of two or one half results in small changes in standard deviations around TCCON (+0.22 ppm and -0.08 ppm, respectively). Changing scattering layer height priors to  $z_{par}=1000$  m or  $z_{par}=5000$  m increased scatter around TCCON by +0.04 ppm and +0.43 ppm, respectively. Similarly, scatter around TCCON changes by +0.22 ppm and -0.05 ppm if  $\alpha_{par}$  is set to 3.0 and 5.0, respectively."

**In the caption of Figure 5:**

added "Comparison of retrieval performances at individual TCCON stations sorted north to south. Marker size indicates amount of colocated soundings at each station. [...] Right: Scatter around TCCON per station for the native, SWIR-1, and SWIR-2 retrievals. Vertical lines mark the average standard deviations (native: 2.43 ppm, SWIR-1: 3.00 ppm, SWIR-2: 3.28 ppm)."

**Page 12, line 5:**

added "..., while correlation coefficients are 0.90 for both SWIR configurations'.'

**Figure 7:**

Inserted correlation coefficients in the lower right corners of both panels.

**In the caption of Figure 7:**

added "Correlation coefficients are displayed in the lower right corners."

**Page 17, line 17:**

added "..., such as the one recently proposed by Hasekamp et al. (2019), ..."

**Page 17, line 30:**

added "Surface reflectance has been shown to be a central driver in methane retrieval precision by Cusworth et al. (2019)."

**Page 18, line 4 - 7:**

added "Additionally, the SWIR-2 seems better suited for the construction of a cloud filter, because its CO2 bands have very different optical depths. Similar to the cloud filter currently in use for GOSAT measurements, one could retrieve XCO2 from the two SWIR-2 bands individually and filter for discrepancies. This scheme should be tested in the future."

Page 18, line 8 – 9:

added "A forthcoming study addressing these aspects of the proposed sensor is currently under preparation."

Figure 12: Introduced a new color scale

**Spectral Sizing of a Coarse Spectral Resolution Satellite Sensor for** XCO2**

Jonas Simon Wilzewski1,2, Anke Roiger1, Johan Strandgren1, Jochen Landgraf3, Dietrich G. Feist4,1,5, Voltaire A. Velazco6, Nicholas M. Deutscher6, Isamu Morino7, Hirofumi Ohvama7, Yao Té8, Rigel Kivi9, Thorsten Warneke10, Justus Notholt10, Manvendra Dubey11, Ralf Sussmann12, Markus Rettinger12, Frank Hase13, Kei Shiomi14, and André Butz15 1Deutsches 
[revised manuscript text omitted]

- For methane (CH4), which poses similar remote sensing challenges as CO2, it has been demonstrated that a satellite spec-10 trometer operating at coarse spectral resolution ( $\frac{\lambda}{\Delta\lambda}$  of a few hundred) on a single absorption band (around 2.35  $\mu$ m) can achieve successful CH4 hot-spot detection with a ground resolution of 30 m (Thompson et al., 2016). Similar results for CH4 have been reported from aircraft sensors that reach ground pixel sizes on the order of 1-10 m (Dennison et al., 2013; Thorpe et al., 2016a, b; Krings et al., 2018). Dennison et al. (2013) suggested that measuring the 2.0  $\mu$ m CO2 bands with a spectral resolution of 10 nm ( $\frac{\lambda}{\Delta\lambda} \approx 200$ ) enables a space-borne spectrometer design that results in ground resolutions as fine as 60×60 m2.
- 15 Thorpe et al. (2016a) have shown that their airborne AVIRIS-NG instrument exploiting the CO2 absorption bands at 2.0  $\mu$ m at a spectral resolution of roughly 5 nm ( $\frac{\lambda}{\Delta\lambda} \approx 400$ ) enables quantitative retrievals of CO2 in localized emission plumes. Thorpe et al. (2016b) suggested that, for CH4, a spectrometer design with a spectral resolution of 1 nm ( $\frac{\lambda}{\Delta\lambda} \approx 2,000$ ) could provide an optimal trade-off that allows for accurate CH4 quantification while supporting small ground pixels.
- This study is motivated by the margins that coarse spectral resolution offers with respect to improving ground resolution and that single-band configurations offer with respect to deploying a fleet of several low-cost satellites. Fig. 1 schematically illustrates the key advantage of an assumed  $50 \times 50$  m2 ground resolution spectrometer over an instrument with km-scale resolution for point-source observation. If the localized source plume does not fill the satellite's entire ground pixel, the XCO2 enhancement averages with the background concentration field over the satellite pixel. For the example in Fig. 1, this leads to a maximum of 3 ppm enhancement for a satellite sensor with  $2 \times 2$  km2 ground resolution. Shrinking the ground pixels leads
- to larger enhancements in the vicinity of the source, simply because the plume fills a larger portion of the (smaller) pixels. In Fig. 1,  $50 \times 50$  m2 ground resolution delivers 12 ppm enhancement at 2 km downwind distance, plus a sampling of the plume cross-section by more than 10 pixels. Further downwind, where the plume has laterally spread to the km-scale, enhancements per pixel are similar for fine and coarse ground resolution, but the fine ground resolution sensor would still sample the plume by multiple ground pixels. Thus, a sensor with fine ground resolution allows for less stringent precision requirements (per ground
- 30 pixel), and it could potentially resolve plume shapes at some detail. Since small ground pixels imply less backscattered photons, sensor design for fine ground resolution typically needs to compensate by enhancing light throughput of the spectrometer and by collecting more photons in the spectral domain, e.g. by coarsening spectral resolution. Since finer ground resolution implies narrower ground coverage for the same detector size, global monitoring with fine ground resolution almost certainly implies the need for a fleet of sensors which would be easier to realize if the sensors had a simple, single-band configuration instead of
- 35 full spectral coverage from the NIR into SWIR-2.

---

## Author Response (AR2)

Manuscript: "Spectral Sizing of a Coarse Spectral Resolution Satellite Sensor for XCO2" by JS Wilzewski et al.

**Reply to anonymous reviewer #4**

We thank the reviewer for the helpful comments to our manuscript. Below we repeat the reviewer's comments in **bold** font and subsequently provide our responses.

**General comments:**

The manuscript presents a study of a potential hypothetical coarse spectral resolution satellite sensor for measuring XCO2 with a ground resolution on the order of 50 x 50 m2. The sensor should be able to discriminate plumes from background concentration fields on the scale of hundreds of meters to a few kilometers. The authors made sensitivity tests with GOSAT cloud free measurements for two SWIR bands and drew conclusions based on the comparisons to the native GOSAT resolution and ground-based reference measurements from the TCCON.

In this study the GOSAT spectra were degraded to lower spectral resolution without further noise being added. As the authors stated, it is important to be aware that their results are only representative under circumstances where the noise can be assumed to be smaller compared to other sources of error. Due to spectral downgrading, the SNR of the coarse resolution spectra is higher as compared to the native GOSAT spectra. However, the effect of measuring at a coarser ILS will also influence the noise and result in lower SNR and this is relevant as the current study is done with significantly higher SNR. The precision calculation shown in this study is valid for the special case where the SNR is very high. Therefore, this has to be clearly mentioned in the conclusions section of the paper. E.g. on page 17 line 1: it would be useful to add – "without further addition of noise to the spectra" for the years 2009 to 2016.

We agree and have adopted this clarification.

Page 1 line 10: please also include the information that no further noise was added while degrading the spectra.

We included this information in the abstract in the sentence: "To this end, we degrade GOSAT SWIR spectra of the  $CO_2$  bands at 1.6 (SWIR-1) and 2.0  $\mu$ m (SWIR-2) to coarse spectral resolution, without a further addition of noise, and we evaluate single-band retrievals of the column-averaged dry-air mole-fractions of CO2 (XCO2) by comparison to..."

**Page 2 line 2: Resolving power of SWIR-1 and SWIR-2 bands are interchanged here – typo?**

Indeed, the resolving powers of the two bands were interchanged in the abstract. Thank you for pointing this out.

Page 5 line 17: Please rephrase the sentence "Since we want..." highlighting the fact that you do not compensate the degradation of the spectral resolution with the addition of the corresponding noise to the spectra but rather keep the same level of noise as in the native high resolution spectra.

We rephrased the sentence

"Since we want to isolate the effects of spectral resolution and spectral band selection, we do not add extra noise to the convolved spectra." to

"Since we want to isolate the effects of spectral resolution and spectral band selection, we do not add extra noise to the convolved spectra i.e. the level of noise is determined by the convolution of the noise of the native GOSAT spectra with the coarse resolution Gaussian line shape function."

**Page 10 line 6: Further information on large biases for some TCCON stations would be useful e.g. Sodankyla, Bremen, Lamont, Edwards**

We do not find the biases for the above TCCON stations to be clearly or systematically increased for the different retrieval configurations that we tested. The station Edwards may be an exception, where a larger bias is observed in all retrievals. However, we cannot really explain this observation. The underlying details and mechanisms that drive the TCCON biases remain inconclusive. Yet, it is worth pointing out that retrieval error sources that depend on spectral resolution, such as spectroscopic uncertainties or aerosol effects, could generally manifest themselves differently in our FP, SWIR-1 and SWIR-2 retrievals as these retrievals all operate at different spectral resolving powers. This may partially explain why biases of certain TCCON stations vary between our FP, SWIR-1 and SWIR-2 setups.

Page 12 line 7: mean difference ("bias") of 0.59 ppm for SWIR-1 and -0.29 ppm for SWIR-2 -> note that the bias values for the SWIR-1 and SWIR-2 are mentioned differently in Fig 9 and 10 - please check or explain why they are different.

Thank you for pointing out this discrepancy. The biases cited on page 12 refer to global biases, which include both land and ocean glint scenes. As we focus on observations over land in this study, we later only subtract the relevant global biases for all land scenes. These are mentioned in Figs. 9 and 10. We have changed the text on page 12 and in the captions of the figures to reflect this.

**Specific comments:**

Page 2 line 5: please switch TCCON-GOSAT by GOSAT-TCCON as the latter is the reference.

Done.

Page 2 line 21: please include the full form of SCIAMACHY Done.

Page 4 line 11: an -> and Done.

Page 18 line 6: "from the two SWIR-2 bands? – why not use the third SWIR-2 band as well?

The third SWIR-2 band near 1.96  $\mu$ m overlaps with strong water vapor absorption lines and therefore it is not well suited for XCO2 retrievals.

**List of Relevant Changes**

Page 1, line 11 – 12:

added ", without a further addition of noise, '

Page 2, line 2:

changed "1,600 (SWIR-1) and 1,200 (SWIR-2)" to "1,200 (SWIR-1) and 1,600 (SWIR-2)"  $\,$

Page 2, line 5: changed "TCCON-GOSAT" to "GOSAT-TCCON"

Page 2, line 9: corrected resolving powers

Page 2, line 21: Spelled out SCIAMACHY

Page 5, line 18-19:

changed "Since we want to isolate the effects of spectral resolution and spectral band selection, we do not add extra noise to the convolved spectra." to "Since we want to isolate the effects of spectral resolution and spectral band selection, we do not add extra noise to the convolved spectra, i.e. the level of noise is determined by the convolution of the noise of the native GOSAT spectra with the coarse resolution Gaussian line shape function."

Page 6, Fig. 2: corrected unit of radiance from W/sr/cm2/cm to W/sr/cm2/cm-1

Page 12, line 9: added "and even glint spectra"

Page 15, Figs. 9,10:

Added "The global mean difference over land ... was subtracted..."

Page 17, line 4:

changed "Performance was evaluated by mimicking the SWIR-1 and SWIR-2 sensors using spectrally degraded GOSAT observations for the years 2009 to 2016..." to "Performance was evaluated by mimicking the SWIR-1 and SWIR-2 sensors using spectrally degraded GOSAT observations without further addition of noise to the spectra for the years 2009 to 2016..."

**Spectral Sizing of a Coarse Spectral Resolution Satellite Sensor for** XCO2**

Jonas Simon Wilzewski1,2, Anke Roiger1, Johan Strandgren1, Jochen Landgraf3, Dietrich G. Feist4,1,5, Voltaire A. Velazco6, Nicholas M. Deutscher6, Isamu Morino7, Hirofumi Ohvama7, Yao Té8, Rigel Kivi9, Thorsten Warneke10, Justus Notholt10, Manvendra Dubey11, Ralf Sussmann12, Markus Rettinger12, Frank Hase13, Kei Shiomi14, and André Butz15 1Deutsches 
[revised manuscript text omitted]

- For methane (CH4), which poses similar remote sensing challenges as CO2, it has been demonstrated that a satellite spec-10 trometer operating at coarse spectral resolution ( $\frac{\lambda}{\Delta\lambda}$  of a few hundred) on a single absorption band (around 2.35  $\mu$ m) can achieve successful CH4 hot-spot detection with a ground resolution of 30 m (Thompson et al., 2016). Similar results for CH4 have been reported from aircraft sensors that reach ground pixel sizes on the order of 1-10 m (Dennison et al., 2013; Thorpe et al., 2016a, b; Krings et al., 2018). Dennison et al. (2013) suggested that measuring the 2.0  $\mu$ m CO2 bands with a spectral resolution of 10 nm ( $\frac{\lambda}{\Delta\lambda} \approx 200$ ) enables a space-borne spectrometer design that results in ground resolutions as fine as 60×60 m2.
- 15 Thorpe et al. (2016a) have shown that their airborne AVIRIS-NG instrument exploiting the CO2 absorption bands at 2.0  $\mu$ m at a spectral resolution of roughly 5 nm ( $\frac{\lambda}{\Delta\lambda} \approx 400$ ) enables quantitative retrievals of CO2 in localized emission plumes. Thorpe et al. (2016b) suggested that, for CH4, a spectrometer design with a spectral resolution of 1 nm ( $\frac{\lambda}{\Delta\lambda} \approx 2,000$ ) could provide an optimal trade-off that allows for accurate CH4 quantification while supporting small ground pixels.
- This study is motivated by the margins that coarse spectral resolution offers with respect to improving ground resolution and that single-band configurations offer with respect to deploying a fleet of several low-cost satellites. Fig. 1 schematically illustrates the key advantage of an assumed  $50 \times 50$  m2 ground resolution spectrometer over an instrument with km-scale resolution for point-source observation. If the localized source plume does not fill the satellite's entire ground pixel, the XCO2 enhancement averages with the background concentration field over the satellite pixel. For the example in Fig. 1, this leads to a maximum of 3 ppm enhancement for a satellite sensor with  $2 \times 2$  km2 ground resolution. Shrinking the ground pixels leads
- to larger enhancements in the vicinity of the source, simply because the plume fills a larger portion of the (smaller) pixels. In Fig. 1,  $50 \times 50$  m2 ground resolution delivers 12 ppm enhancement at 2 km downwind distance, plus a sampling of the plume cross-section by more than 10 pixels. Further downwind, where the plume has laterally spread to the km-scale, enhancements per pixel are similar for fine and coarse ground resolution, but the fine ground resolution sensor would still sample the plume by multiple ground pixels. Thus, a sensor with fine ground resolution allows for less stringent precision requirements (per ground
- 30 pixel), and it could potentially resolve plume shapes at some detail. Since small ground pixels imply less backscattered photons, sensor design for fine ground resolution typically needs to compensate by enhancing light throughput of the spectrometer and by collecting more photons in the spectral domain, e.g. by coarsening spectral resolution. Since finer ground resolution implies narrower ground coverage for the same detector size, global monitoring with fine ground resolution almost certainly implies the need for a fleet of sensors which would be easier to realize if the sensors had a simple, single-band configuration instead of
- 35 full spectral coverage from the NIR into SWIR-2.

**Figure 1.** Schematic Gaussian plume of the XCO2 enhancement ( $\Delta$ XCO2) originating from a power plant with 12.3 Mt CO2 
[revised manuscript text omitted]